# ATRX promotes heterochromatin formation to protect cells from G-quadruplex DNA-mediated stress

Yu-Ching Teng[1], Aishwarya Sundaresan[1], Ryan O'Hara[1], Vincent U. Gant[1], Minhua Li[1], Sara Martire [1], Jane N. Warshaw[1], Amrita Basu[2] & Laura A. Banaszynski [1✉]

ATRX is a tumor suppressor that has been associated with protection from DNA replication stress, purportedly through resolution of difficult-to-replicate G-quadruplex (G4) DNA structures. While several studies demonstrate that loss of ATRX sensitizes cells to chemical stabilizers of G4 structures, the molecular function of ATRX at G4 regions during replication remains unknown. Here, we demonstrate that ATRX associates with a number of the MCM replication complex subunits and that loss of ATRX leads to G4 structure accumulation at newly synthesized DNA. We show that both the helicase domain of ATRX and its H3.3 chaperone function are required to protect cells from G4-induced replicative stress. Furthermore, these activities are upstream of heterochromatin formation mediated by the histone methyltransferase, ESET, which is the critical molecular event that protects cells from G4-mediated stress. In support, tumors carrying mutations in either *ATRX* or *ESET* show increased mutation burden at G4-enriched DNA sequences. Overall, our study provides new insights into mechanisms by which ATRX promotes genome stability with important implications for understanding impacts of its loss on human disease.

[1] Cecil H. and Ida Green Center for Reproductive Biology Sciences, Department of Obstetrics and Gynecology, Children's Medical Center Research Institute, Harold. C. Simmons Comprehensive Cancer Center, Hamon Center for Regenerative Science and Medicine, University of Texas Southwestern Medical Center, Dallas, TX, USA. [2] Department of Surgery, University of California San Francisco, San Francisco, CA, USA. ✉email: Laura. Banaszynski@UTSouthwetsern.edu

ATRX is a chromatin remodeling protein of the SWI/SNF family, mutations in which cause alpha thalassemia X-linked (ATRX) intellectual disability syndrome and are highly associated with a number of cancers characterized by alternative lengthening of telomeres (ALT)[1]. ATRX localizes to heterochromatic repetitive regions, including telomeres, pericentric repeats, rDNA repeats, and endogenous retroviral elements (ERVs)[2–8]. ATRX functions with the histone chaperone DAXX to facilitate deposition of the histone variant H3.3 at these regions[9], resulting in a unique form of heterochromatin characterized by both trimethylation of histone H3 at lysine 9 (H3K9me3) and H3.3[10,11]. Loss of any one of ATRX/DAXX/H3.3 is correlated with loss of H3K9me3 heterochromatin at repetitive regions[3,11,12]. Loss of ATRX has been implicated in replication stress, DNA damage, and DNA repair failures that drive genome instability, and aberrant homologous recombination[6,8,12–18]. However, the mechanism underlying this dysfunction is unclear.

ATRX is hypothesized to prevent replication stress by resolving stable non-B form DNA structures called G-quadruplexes (G4) ahead of the replication fork. These structures are thought to form when double-stranded DNA is dissociated in GC-rich regions during replication and transcription[19,20]. G4 structures are generally considered to block the progression of DNA replication forks, ultimately leading to replication fork collapse[21]. A number of helicases, including BLM, WRN, and ATRX, are proposed to protect the genome by unfolding G4 structures and preventing them from causing DNA breaks[22–24]. Genome-wide, ATRX is enriched at GC-rich sequences with the propensity for forming G4 structures[2] although ATRX is unable to unfold G4 structures in vitro[14]. However, exogenous expression of ATRX in cells lacking ATRX has been shown to reduce levels of G4 structures[25] and, further, ATRX has been shown to protect cells from replication stresses induced by chemical stabilizers of G4 structures[15,25,26]. Despite these striking observations, direct links between ATRX, G4 structures, and the molecular mechanisms by which ATRX functions at these regions have not been reported.

In this study we explored the mechanisms underlying ATRX function at G4 structures. We validate previous observations that ATRX is bound at G4-containing regions and further demonstrate that ATRX interacts with G4 structures in cells. In addition, we find that ATRX interacts with the MCM helicase and that G4 DNA accumulates at newly synthesized DNA in the absence of ATRX. Genomic analysis suggests that loss of ATRX results in G4 accumulation primarily at intergenic and repetitive regions. We show that ATRX requires both its helicase and H3.3 chaperone activity to attenuate G4 stabilization-induced replication stress. Interestingly, we find that ATRX maintains a closed chromatin state at G4-containing regions and that this function is mediated by DAXX-dependent deposition of H3.3. These activities are upstream of heterochromatin formation, which is ultimately required for protection from G4-mediated stress. Finally, we show that pan-cancer patient samples bearing mutations in either ATRX or ESET had a significantly increased mutation burden at G4-containing DNA. Overall, our findings support the conclusion that ATRX protects genomic integrity at G4-containing regions by maintaining these regions in a closed heterochromatic state.

## Results

**ATRX associates with G4 structures in vivo.** It has been reported that ATRX binds to repetitive regions of the genome such as telomeres and pericentric regions[2], but also to promoters[7,8]. To confirm genomic ATRX enrichment, we performed ChIP-seq using a GFP antibody and mouse embryonic stem cells (mESCs) in which the endogenous ATRX locus was tagged[27] with a C-terminal GFP (Fig. 1a and Supplementary Fig. 1a, b). ATRX-enriched regions were often enriched with the histone variant, H3.3, and with H3K9me3, consistent with previous reports of this unique class of heterochromatin in ESCs[11,28] (Fig. 1a). In agreement with published data[2,8], we found ATRX enriched mainly at repeat regions in ESCs, including telomere and pericentric repeats that are predicted to form G-quadruplex structures (G4) under physiological conditions[29,30] (Supplementary Fig. 1c–e). To further validate these results, we performed ChIP-seq using an ATRX antibody in both WT and ATRX knockout (ATRX KO) ESCs (Fig. 1a and Supplementary Fig. 1f, g). For further analysis, we considered our ATRX-GFP and WT ATRX ChIP-seq data sets as replicates. Overall, we identified 435 regions of ATRX enrichment at uniquely mappable regions of the genome. These regions contained 3.6× more (30 standard deviations more than expected by chance) G4 consensus motifs $(G_3N_{1-12}G_3N_{1-12}G_3N_{1-12}G_3)$[31] compared to randomly selected regions that were comparable in size and number, over 100 simulations (Supplementary Fig. 1h). Nearly 40% (169/435) of ATRX-enriched regions contained a G4 consensus motif (Fig. 1b and Supplementary Data File 1), and further, de novo motif prediction identified GC-rich motifs as being over-represented in these regions (Supplementary Fig. 1i). G4-containing regions showed considerable ATRX enrichment compared to regions identified as enriched with ATRX but without a G4 consensus motif (Fig. 1c and Supplementary Fig. 1j, k).

Based on the enrichment of the G4 consensus motif at sites of ATRX enrichment, we next asked whether we could observe ATRX interaction with G4 structures in cells. Since ATRX is reported to be recruited to telomeres during late S and associated with their replication[13], we wanted to assess ATRX-G4 interaction throughout the cell cycle. We synchronized cells in mitosis using a thymidine-nocodazole treatment and then released cells with EdU to visualize DNA replication (Fig. 1d and Supplementary Fig. 2a, b). We then performed a proximity ligation assay (PLA) using antibodies that recognize 1) ATRX and 2) G4 structures (e.g., the BG4 antibody)[32] in both WT and ATRX KO mESCs. Interestingly, we observed ATRX and G4 association in cells that had not yet begun replication (Fig. 1e, note cells in G1 evidenced by lack of EdU staining). Association was highest in G1 phase and decreased slightly in early S phase (Fig. 1f). We did not observe an appreciable level of ATRX-G4 association on mitotic chromosomes (Supplementary Fig. 2c). Further, we did not observe ATRX-G4 foci in single antibody controls or ATRX KO mESCs (Supplementary Fig. 3a) or G4 association with an independent chromatin-associated protein, HIRA (Supplementary Fig. 3b, c), demonstrating the specificity of the interaction. Overall, these data strongly link ATRX to regions containing G4 consensus motifs and G4 structures.

**ATRX associates with the MCM DNA helicases.** Despite increasing evidence that ATRX plays a replication-associated role in resolving G4 structures[14–16], mechanisms that recruit ATRX to these structures remain poorly understood. We therefore used immunoprecipitation-coupled to mass spectrometry (IP-MS) to identify potential factors that interact with ATRX that may allow insight into its localization and function. We identified many proteins related to DNA replication and transcription that were enriched by ATRX pulldown in wild-type cells compared to ATRX KO cells and immunoprecipitation using a control IgG antibody (Fig. 2a and Supplementary Fig. 4a, b). These include previously identified ATRX-interacting proteins such as Mre11[14], a member of the MRN complex involved in DNA damage repair (Supplementary Data File 2). Further, we identified a number of

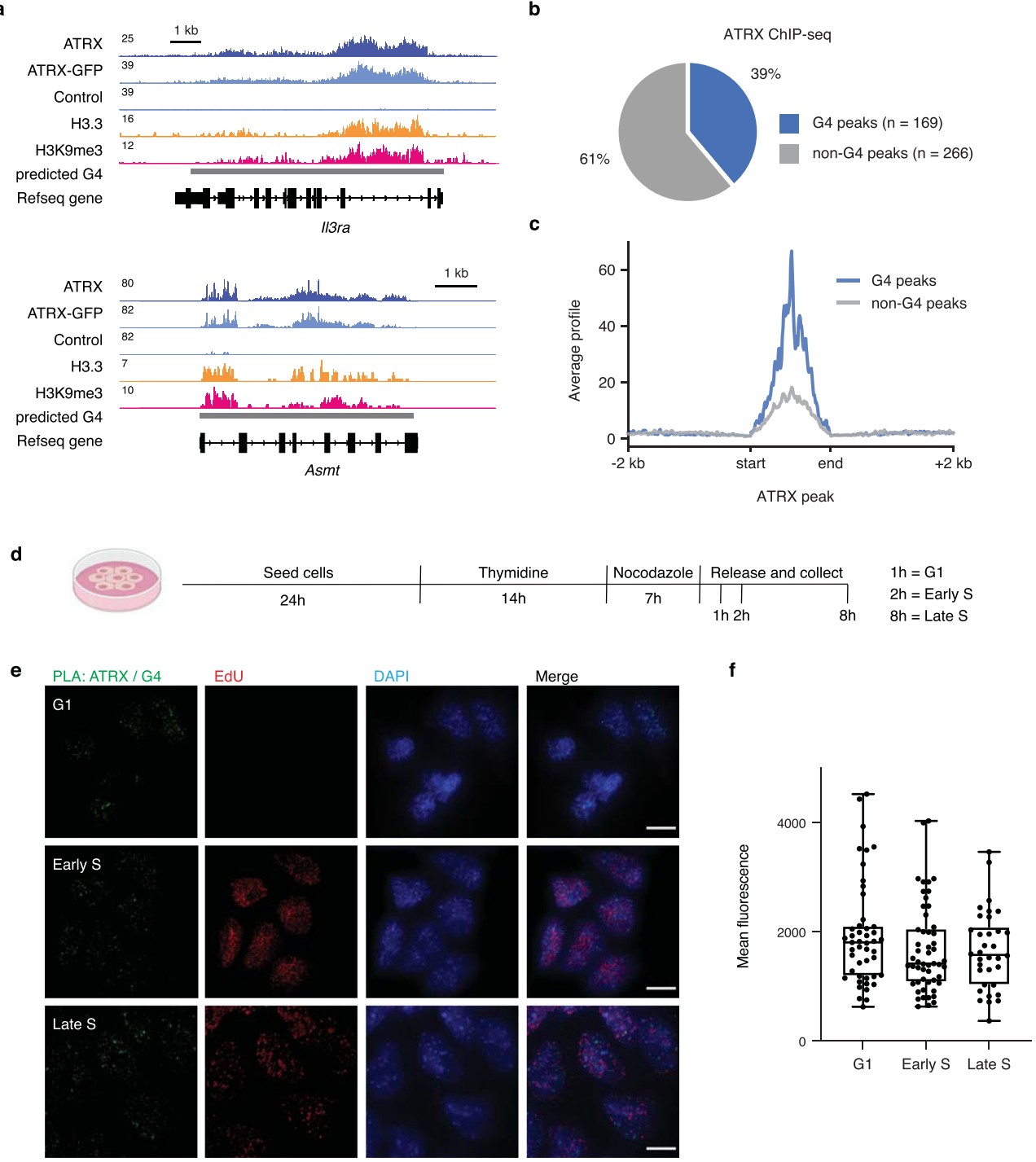

**Fig. 1 ATRX associates with G4 structures in vivo. a** Genome browser representations of ATRX, ATRX-GFP, H3.3, and H3K9me3 ChIP-seq at predicted G4 regions in ESCs. Data represented as read density in reads per kilobase per million mapped reads (RPKM) normalized to an external standard for each data set. Gray boxes indicate predicted G4 regions. **b** ChIP-seq analysis of ATRX enrichment in ESCs. Pie chart represents the percentage of ATRX-enriched regions containing G4 consensus motifs (169/435, 39%). **c** ATRX ChIP-seq average profiles in ESCs at ATRX-enriched regions containing G4 consensus motifs (G4) compared with ATRX-enriched regions that do not contain a G4 consensus motif (non-G4). **d** Schematic of ESCs synchronization protocol. Cells are incubated with thymidine for 14 h, washed, and treated in medium with nocodazole for 7 h. After washing, mitotic cells are released in medium and incubated with EdU in prior cell fixation for downstream experiments. Cells in G1, early S, and late S phase were analyzed 1, 2, and 8 h after release, respectively. **e** Representative images demonstrating ATRX and G4 co-localization by proximity ligation assay (PLA) in synchronized ESCs. Green—PLA (ATRX-G4). Red—EdU-labeling, indicative of newly synthesized DNA. Blue—DAPI nuclear stain. Scale bar equals 10 μm ($n = 1$ independent experiment). **f** Quantification of signal intensity from ATRX-G4 PLA foci in G1 ($n = 47$), early S ($n = 53$), and late S ($n = 34$) phases of ESCs. Quantification of PLA data are presented as box-and-whisker plots marking a horizontal median line. The bottom and the top of the boxes correspond to the 25 and 75th percentiles, and the internal band is the 50th percentile (median). The plot whiskers show down to the minimum and up to the maximum value. One-way ANOVA, $p = 0.2554$ (G1 vs. early S), $p = 0.2760$ (G1 vs. late S), and $p = 0.9905$ (early S vs. late S).

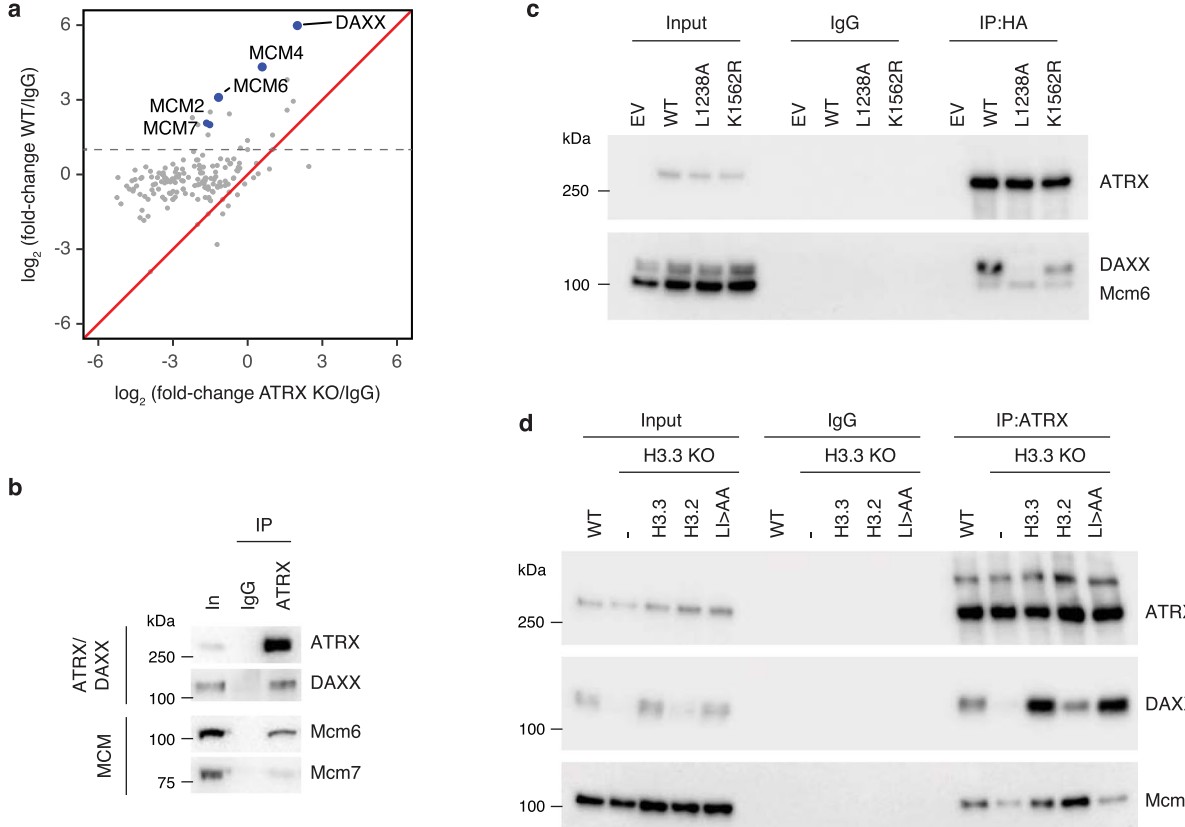

**Fig. 2 ATRX associates with the MCM DNA helicase. a** Proteomic analysis of ATRX-interacting proteins. Mass spectrometry data represented a scatter plot of the log2 abundance ratios of ATRX-enriched proteins compared to IgG control in both wild-type and ATRX KO HeLa cells. Gray dotted line indicates a fold-change >2 in ATRX-enriched proteins compared to IgG control. ($n = 1$ independent experiment). **b** Co-immunoprecipitation from wild-type ESC nuclear extracts showing ATRX interaction with DAXX, Mcm6 and Mcm7. **c** Co-immunoprecipitation from ATRX KO addback ESC nuclear extracts showing HA-tagged ATRX interaction with DAXX and Mcm6. K1562R—ATRX mutation in helicase domain; L1238A—ATRX mutation in DAXX-binding motif. **d** Co-immunoprecipitation from H3.3 KO addback ESC nuclear extracts showing ATRX interaction with DAXX and Mcm6. H3.3 LI/AA—H3.3 deposition mutant. For (**b**–**d**), data are representative of experiments performed at least 3 times. For all immunoblots, source data are provided as a Source Data file.

proteins that have been genetically linked to G4 structures[33], including DDX10 and SPEN (Supplementary Data File 2). We also identified members of the MiniChromosome Maintenance (MCM) complex (e.g., MCM2, MCM4, MCM6, and MCM7), a DNA helicase essential for replication[34], as novel ATRX-interacting proteins (Fig. 2a).

We first validated interaction of ATRX and MCM proteins using co-immunoprecipitation (Fig. 2b and Supplementary Fig. 4c). ATRX specifically pulled down MCM2/4/6/7 subcomplexes[34,35] (Fig. 2a, b and Supplementary Fig. 4c). ATRX interaction with Mcm6 persists in lysates treated with either ethidium bromide or RNase A (Supplementary Fig. 4d). Because the enrichment of MCM by ATRX immunoprecipitation was relatively low, and to determine whether ATRX-MCM interaction occurred in cells, we performed proximity ligation using ATRX and MCM antibodies. We observed ATRX interaction with MCM2/3/4/6, with no PLA signal observed in ATRX KO ESCs (Supplementary Fig. 4e–l), suggesting that ATRX may be in proximity of a complete MCM complex. The majority of EdU labeling is not at sites of ATRX-MCM interaction, in line with ATRX being bound at only a small number of loci genome-wide (Fig. 1b).

ATRX contains an ATPase/helicase domain that is often mutated in both ATRX syndrome and in cancers[36,37]. Another well-characterized function of ATRX is its involvement in H3.3 deposition at telomeres and other types of heterochromatin

through its interactions with the H3.3 chaperone protein, DAXX[9]. Hypothetically, the ATRX helicase domain could unwind G4 structures which could be resolved by H3.3 deposition and nucleosome formation. We therefore asked whether interaction with MCM requires these two functional elements of ATRX. We first performed ATRX immunoprecipitation and ATRX-Mcm6 PLA in DAXX knockout (DAXX KO) ESCs (Supplementary Fig. 1f). Interestingly, we found that ATRX maintains association with Mcm6 in the absence of DAXX using both IP-WB and PLA (Supplementary Fig. 5a–c). In support, the ATRX-Mcm6 association was maintained in ATRX KO cells expressing an ATRX mutant that attenuates DAXX-binding (ATRX L1238A)[38] (Fig. 2c and Supplementary Fig. 5d–f). The ATRX/DAXX chaperone complex shares its substrate H3.3 with the HIRA complex, which deposits H3.3 at regulatory elements and genes[39]. We found that HIRA did not co-immunoprecipitate MCM6 in wild-type cells (Supplementary Fig. 5g), and that ATRX interaction with Mcm6 was maintained in the absence of HIRA (Supplementary Fig. 5a, h), suggesting the specificity of the ATRX-MCM interaction. Finally, we expressed an ATRX helicase mutant (ATRX K1562R) in ATRX KO mESCs. Using IP-WB and PLA, we found that the ATRX helicase mutant maintains association with Mcm6 (Fig. 2c and Supplementary Fig. 5d–f). Overall, these results suggest that ATRX helicase and chaperone activity is decoupled from mechanisms dictating its localization on chromatin.

Recent reports demonstrate that Mcm2 itself contains chaperone function for H3-H4 dimers[40], raising the intriguing possibility that the MCM-ATRX association is bridged by H3.3 itself. To test the requirement of H3.3 for ATRX-Mcm6 interaction, we immunoprecipitated ATRX from H3.3 knockout cells (Supplementary Fig. 6a). We found that loss of H3.3 resulted in reduced association between ATRX and Mcm6 (Fig. 2d and Supplementary Fig. 6b). ATRX-Mcm6 association could be restored by exogenous expression of H3.3 but not by an H3.3 mutant previously shown to inhibit stable formation of nucleosomes (H3.3 L126A I130A)[41] (Fig. 2d and Supplementary Fig. 6c), demonstrating the importance of H3.3 deposition for maintaining the ATRX-MCM interaction.

**Loss of ATRX leads to accumulation of G4 structures at sites of DNA synthesis.** Our identification of an association between ATRX and the MCM complex supports a body of existing literature demonstrating that ATRX plays a role in DNA replication[14–16,26,42]. While many studies hypothesize that ATRX replicative stress is due to the inability to resolve G4 structures[2,14,15,25,26], the direct effect of loss of ATRX on G4 structures at replicating DNA has not been assessed. To test whether the persistence of G4 structures at newly synthesized DNA is affected by loss of ATRX, we used in situ analysis of protein interaction at DNA replication forks, or SIRF[43]. Briefly, mitotic cells were released from synchronization and labeled with EdU for a short period of time prior to cell fixation at early S phase. EdU incorporation into newly synthesized DNA was detected using click chemistry for biotin labeling. EdU and G4 interaction was then determined by proximity ligation. We found that normalized EdU-G4 signals were increased in ATRX KO and DAXX KO ESCs (Fig. 3a, b). This effect could be rescued by expression of wild-type ATRX in ATRX KO ESCs, whereas we find that both the helicase activity of ATRX and its ability to interact with DAXX are important to reduce G4-EdU persistence (Supplementary Fig. 7).

While imaging experiments allow powerful observation of G4 activity at the single-cell level, they fail to provide genomic identification of specific G4-containing regions that may be affected by loss of ATRX. To experimentally identify G4 regions in ECSs and determine the effect of ATRX loss on these regions, we performed CUT&Tag using the BG4 antibody[44] (Fig. 3c). We identified 2320 regions as experimentally enriched with G4 structures in wild-type cells (Fig. 3d). These regions contained 5.3× more G4 motifs compared to randomly selected regions, with 58% (1356/2230) containing a predicted G4 motif (Supplementary Fig. 8a, b). Generally, G4-containing regions are localized to promoters in wild-type ESCs—of note, a region that we do not strongly associate with ATRX binding (Fig. 3e, also see Supplementary Fig. 1c). We next wanted to determine the effect of loss of ATRX on genomic G4 accumulation. Interestingly, we observe 6301 novel G4 sites in ATRX KO ESCs that are not observed in wild-type ESCs (Fig. 3d and Supplementary Fig. 8c, d). These regions are annotated predominantly as introns, intergenic regions, LINEs, and LTRs (Fig. 3f), suggesting that ATRX is mainly responsible for resolving G4 structures in these regions.

While the G4-EdU signal observed in our PLA assay could be due to replication, we cannot rule out that this signal results from nucleotide incorporation due to DNA damage repair. To try to distinguish between the two, we used two sequencing techniques—EdU-seq, in which we sequenced nascently synthesized DNA after a short pulse of EdU (which cannot distinguish between repair and replication), and published SNS-seq data[45], in which RNA primers associated with DNA replication initiation are sequenced (mapping

putative origins). We found several regions where mapped origins[45] were identified upstream or downstream of ATRX-enriched G4 regions, and further, that treatment with the G4 stabilizer, PhenDC3, resulted in increased detection of origin activity at or near ATRX-bound G4 regions compared with untreated cells (Supplementary Fig. 9a–d). These regions also showed increased EdU-seq signal in the absence of either ATRX or DAXX (Supplementary Fig. 9a, e, f). However, when we consider G4-enriched regions that are identified experimentally in the absence of ATRX, we find that these regions show increased EdU-seq signal in the absence of ATRX or DAXX with no evident increase in origin usage after PhenDC3 treatment[45] (Fig. 3g and Supplementary Fig. 9g). Overall, while a subset of ATRX-dependent G4 regions may stimulate replication, these data and the literature are consistent with a model in which ATRX KO cells experience increased replicative stress and DNA damage due to G4 persistence.

**ATRX requires its helicase activity and interaction with DAXX to protect from G4-induced stress.** Previous studies have linked loss of ATRX to increased sensitivity to chemical agents that stabilize G4 structures such as pyridostatin (PDS) and CX-3543, presumably due to G4 persistence in cells[25,26]. Our results suggest that both ATRX and DAXX are required to reduce G4 levels in cells. We therefore asked whether loss of DAXX would phenocopy loss of ATRX with respect to PDS sensitivity. We treated an isogenic panel of wild-type, ATRX KO, and DAXX KO ESCs with increasing concentrations of PDS. Analysis of cell viability demonstrated that both DAXX KO and ATRX KO resulted in increased sensitivity to PDS (Fig. 4a). Importantly, exogenous expression of DAXX in DAXX KO ESCs, and likewise, exogenous expression of ATRX in ATRX KO ESCs, attenuated sensitivity to PDS (Fig. 4b and Supplementary Fig. 10a, b). To assess whether ATRX interaction with DAXX was necessary to relieve PDS-induced stress, we expressed the ATRX DAXX-binding mutant (ATRX L1238A) in ATRX KO ESCs. Strikingly, this mutation is unable to relieve PDS sensitivity (Fig. 4b), suggesting that ATRX interaction with DAXX is required for this function. Further, mutations to ATRX that disrupt its helicase activity (ATRX K1562R, K1612N)[46] remain sensitive to PDS (Fig. 4b).

The requirement for an ATRX/DAXX complex suggests that H3.3 deposition may be important to protect from PDS sensitivity. In agreement, we found that loss of either ATRX or DAXX results in reduced H3.3 incorporation at sites of ATRX enrichment containing G4 consensus motifs (Supplementary Fig. 10c, d). Further, H3.3 KO ESCs showed increased sensitivity to PDS compared to wild-type ESCs (Fig. 4c and Supplementary Fig. 10e). Sensitivity could be rescued by exogenous expression of H3.3 but not by the H3.3 deposition mutant (H3.3 L126A I130A) (Fig. 4c). Importantly, we find that HIRA KO ESCs are highly comparable with wild-type ESCs with respect to PDS sensitivity (Fig. 4a). Overall, these data suggest that both ATRX helicase activity and ATRX/DAXX-mediated H3.3 deposition, but not HIRA-mediated H3.3 deposition, are required for protection from G4 stabilizers.

**ATRX maintains closed chromatin states upstream of heterochromatin formation to protect cells from G4-mediated stress.** We next wanted to understand how ATRX protects cells from G4-mediated stress. If ATRX is involved in nucleosome assembly at G4 DNA, it follows that these regions should be more open in the absence of ATRX. In support, recent studies demonstrate that loss of ATRX induces chromatin de-compaction at telomeres and repetitive elements[3,10,12,47]. To explore whether ATRX deficiency altered chromatin accessibility at G4 regions, we performed ATAC-seq in wild-type and ATRX KO ESCs. Strikingly, we

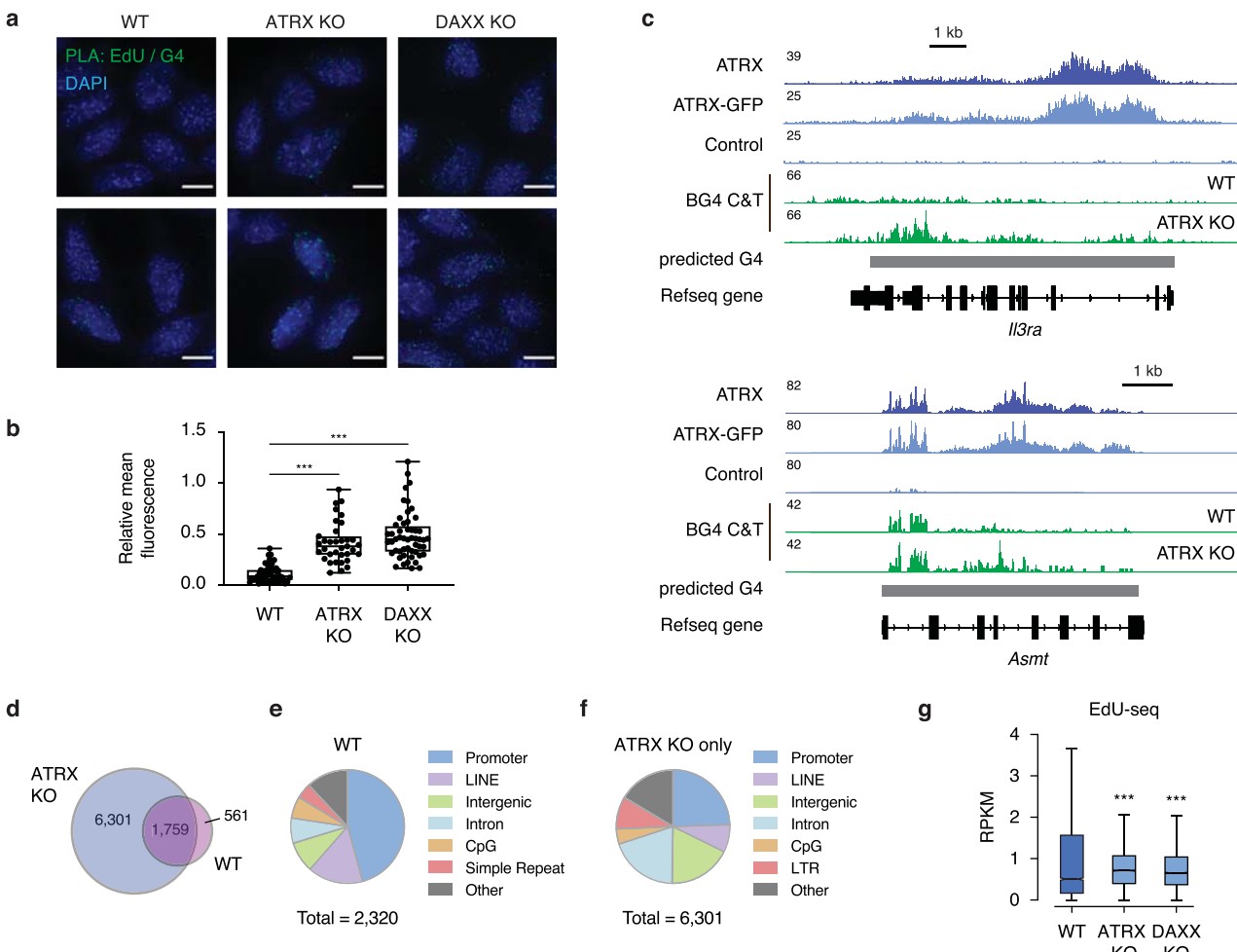

**Fig. 3 ATRX prevents the accumulation of G4 structures at sites of DNA synthesis. a** Representative images demonstrating EdU and G4 co-localization by proximity ligation assay (PLA) in synchronized ESCs at early S phase. Green—PLA (EdU-G4). Blue—DAPI nuclear stain. Scale bar equals 10 μm ($n = 2$ independent experiments). **b** Quantification of signal intensity from EdU-G4 PLA foci in wild-type ($n = 48$), ATRX KO ($n = 35$) ($p = 2 \times 10^{-10}$) and DAXX KO ($n = 52$) ($p = 1 \times 10^{-10}$) ESCs. The bottom and the top of the boxes correspond to the 25th and 75th percentiles, and the internal band is the 50th percentile (median). The plot whiskers show down to the minimum and up to the maximum value. Statistical significance determined by a One-way ANOVA test. ***$p < 0.001$. **c** Genome browser representations of ATRX, ATRX-GFP ChIP-seq, and BG4 CUT&Tag at observed G4 regions in ESCs. **d**–**f** CUT&Tag analysis of G4 enrichment in ESCs. **d** Venn diagram showing the unique and overlaps between wild-type and ATRX KO ESCs. **e**, **f** Pie chart of genomic region annotation in wild-type (**e**) and ATRX KO ESCs (**f**). **g** Box plots representing EdU-seq read counts of early S phase in wild-type, ATRX KO ($p = 8.9 \times 10^{-11}$) and DAXX KO ($p = 1.6 \times 10^{-6}$) ESCs at observed G4 regions that are only identified in ATRX KO ($n = 6301$). Data are representative of two independent experiments. The bottom and the top of the boxes correspond to the 25 and 75th percentiles, and the internal band is the 50th percentile (median). The plot whiskers correspond to 1.5 interquartile range. Statistical significance determined by Wilcoxon–Mann–Whitney test. ***$p < 0.001$.

found that ATRX depletion elevated chromatin accessibility at ATRX-enriched regions containing G4 consensus motifs while paradoxically resulting in reduced H3.3 deposition, a histone variant long associated with open chromatin (Fig. 5a, b and Supplementary Fig. 10c). Analogous results were observed in DAXX KO ESCs and H3.3 KO ESCs (Fig. 5c, d and Supplementary Fig. 10d). Further, we find that expression of exogenous ATRX in ATRX KO ESCs rescues this effect, while neither the ATRX helicase mutant nor the DAXX-interaction mutant in ATRX KO ESCs results in reduced accessibility at these regions (Fig. 5e). Maintaining these regions in a closed state requires H3.3 deposition, as only expression of wild-type H3.3, and not that of the deposition mutant, was able to rescue this effect in H3.3 KO ESCs (Fig. 5f). Importantly, chromatin accessibility at G4 regions was not altered in HIRA KO ESCs (Supplementary Fig. 11a). Additionally, increased chromatin accessibility did not result in concomitant increases in histone acetylation or transcription from nearby genes (Supplementary Fig. 11b–f).

Previous studies have shown that ATRX facilitates the establishment and maintenance of a subset of H3.3- and H3K9me3-marked heterochromatin[3,5,11]. We therefore asked how loss of the ATRX/DAXX/H3.3 complex would influence the chromatin state at ATRX-enriched G4 regions. First, we re-analyzed published H3K9me3 ChIP-seq data for ATRX KO, DAXX KO and H3.3 KO ESCs[11]. We found that H3K9me3 enrichment at ATRX-bound G4 regions was reduced in ATRX KO, DAXX KO, and H3.3 KO cells (Fig. 6a–d). Further, we find that expression of exogenous ATRX in ATRX KO ESCs rescues this effect, while neither the ATRX helicase mutant nor the DAXX-interaction mutant in ATRX KO ESCs rescues heterochromatin formation at these regions (Fig. 6e). Maintaining these regions in a closed state requires H3.3 deposition, as only expression of wild-type H3.3, and not that of the deposition mutant, was able to rescue this effect in H3.3 KO ESCs (Fig. 6f). These data demonstrate that the ATRX/DAXX/H3.3 pathway are key factors for H3K9me3 maintenance at G-quadruplex regions.

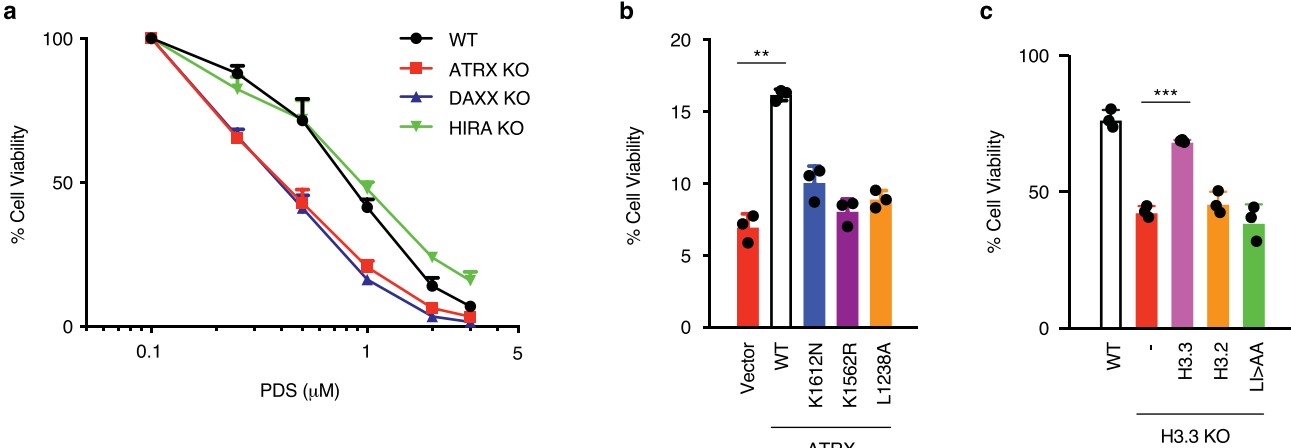

**Fig. 4 ATRX requires its helicase and chaperone activity to protect from G4-induced stress. a** Cell viability of wild-type, ATRX KO, DAXX KO, and HIRA KO ESCs treated with PDS for 5 days. IC50: wild-type—0.804 μM, ATRX KO—0.396 μM, DAXX KO—0.384 μM, and HIRA KO—0.909 μM. **b** Cell viability of ATRX KO ESCs exogenously expressing either wild-type or mutant ATRX constructs treated with 2 μM PDS for 5 days. K1612N and K1562R—ATRX mutations in helicase domain; L1238A—ATRX mutation in DAXX-binding motif. **c** Cell viability of H3.3 KO ESCs exogenously expressing either wild-type or mutant H3.3 (L126A I130A, LI/AA) or H3.2 constructs treated with 1 μM PDS for 5 days. For both panels, mock-treated cells at day 0 were taken as 100% survival. Data represented as mean ± SD ($n = 3$). Statistical significance determined by a Two-way ANOVA test. **$p < 0.01$; ***$p < 0.001$. $p = 7.2 \times 10^{-5}$ (vector vs. WT in **b**), and $p = 1.4 \times 10^{-5}$ (H3.3 KO vs. H3.3 addback in **c**).

Previous studies demonstrate that the histone methyltransferase ESET has a critical role in the establishment of H3K9me3 at H3.3-enriched repetitive elements[11,48]. Accordingly, H3K9me3 enrichment on H3.3-containing nucleosomes at ATRX-bound G4 regions was reduced in the absence of ESET (Fig. 7a, b and Supplementary Fig. 12a). We next asked whether there exists a molecular hierarchy (ATRX helicase and histone chaperone activity vs. ATRX promotion of heterochromatin formation) in protecting cells from G4-mediated stress. To this end, we used ESET conditional KO ESCs to determine the effect of loss of heterochromatin at G4 regions. First, we found that H3.3 deposition at ATRX-bound G4 regions is unaffected in the absence of ESET (Fig. 7c), even though heterochromatin at these regions is reduced (Supplementary Fig. 12b, c). Interestingly, ATRX-bound G4 regions become more open in the absence of ESET (Fig. 7d). Further, like ATRX/DAXX/H3.3 KO ESCs, ESET KO ESCs are sensitive to PDS treatment (Fig. 7e). Taken together, these results suggest that heterochromatin formation is the ultimate molecular event that protects cells from G4-mediated stress.

It has been shown that G4 structures are correlated with increased DNA polymerization error rates[49], and that G4 structures harbor a high incidence of disease-causing point mutations and indels[50–52]. Further, studies demonstrate that ATRX mutant tumors carry a high mutation rate either at the single-nucleotide variant level or copy number alterations[25,53]. Given these observations, we asked whether regions identified as enriched with G4 structures had increased mutation burden in human cancers. Strikingly, we found that patient samples containing mutations in either *ATRX* or *ESET* contained a significantly higher number of substitutions at G4-containing DNA compared to an iteratively sampled random selection of donors (Fig. 8a, b). These substitutions were more likely to involve mutation of G or C bases compared with A or T (Fig. 8c, d). Further, this trend was not observed from donors carrying mutations in oncogenes such as *KRAS* and *IDH1* (Supplementary Fig. 13), lesions that we do not expect to impact G4 DNA regulation.

## Discussion

Previous studies have linked ATRX to replication fork progression, presumably through resolution of G4 structures[2,14,15,25,26]. Here, we provide a direct link between ATRX and the replication machinery. We find that ATRX interacts with the MCM helicase and that G4 structures show increased coincidence with newly synthesized DNA in the absence of ATRX. We find that ATRX maintains G4-containing regions in a closed heterochromatic state through DAXX-mediated deposition of the histone variant H3.3, and that establishment of heterochromatin is the critical molecular event protecting cells from G4-mediated stress (Fig. 9).

Previous studies using immunofluorescence have not observed co-localization of ATRX and G4 structures in cells, presumably because ATRX activity results in the resolution of G4s[25]. One advantage of PLA is the ability to capture these brief interactions. Interestingly, we find that ATRX colocalizes with G4 structures throughout both G1 and S phase. This result suggests that ATRX recruitment to these regions is not a result of replication fork stalling, and that it is instead a normal feature of ATRX function. While we observe co-localization between ATRX and the MCM complex, the exact molecular nature of this interaction remains to be seen and is unlikely to drive ATRX recruitment to chromatin, given that MCM-bound regions vastly outnumber ATRX-bound regions. To date, binding of G4 structures has not been attributed to a particular region of ATRX. ATRX-bound regions thought to contain G4 structures, such as telomeres, are generally enriched with H3K9me3, and ATRX has been shown to bind H3K9me3 through its ADD domain[54,55]. However, ESCs contain many regions of H3K9me3 heterochromatin that are not bound by ATRX or enriched with H3.3[11], so this cannot be the sole determinant of recruitment and instead may stabilize ATRX association with these regions. Interestingly, we do not observe appreciable ATRX enrichment at the thousands of new G4 structures that we identify genomically in the absence of ATRX, in support of the hypothesis that ATRX recruitment to these regions may be transitory and that ATRX is released once the G4 has been resolved. It is also formally possible that ATRX may in some way regulate other helicases responsible for regulating G4 structures in cells. Detailed molecular understanding of the kinetics of ATRX recruitment and function at G4 regions remains an important point of future study.

Strikingly, we observe a dramatic increase in chromatin accessibility at ATRX-enriched G4 regions in the absence of ATRX/DAXX/H3.3, accompanied by a loss of H3K9me3 heterochromatin at these

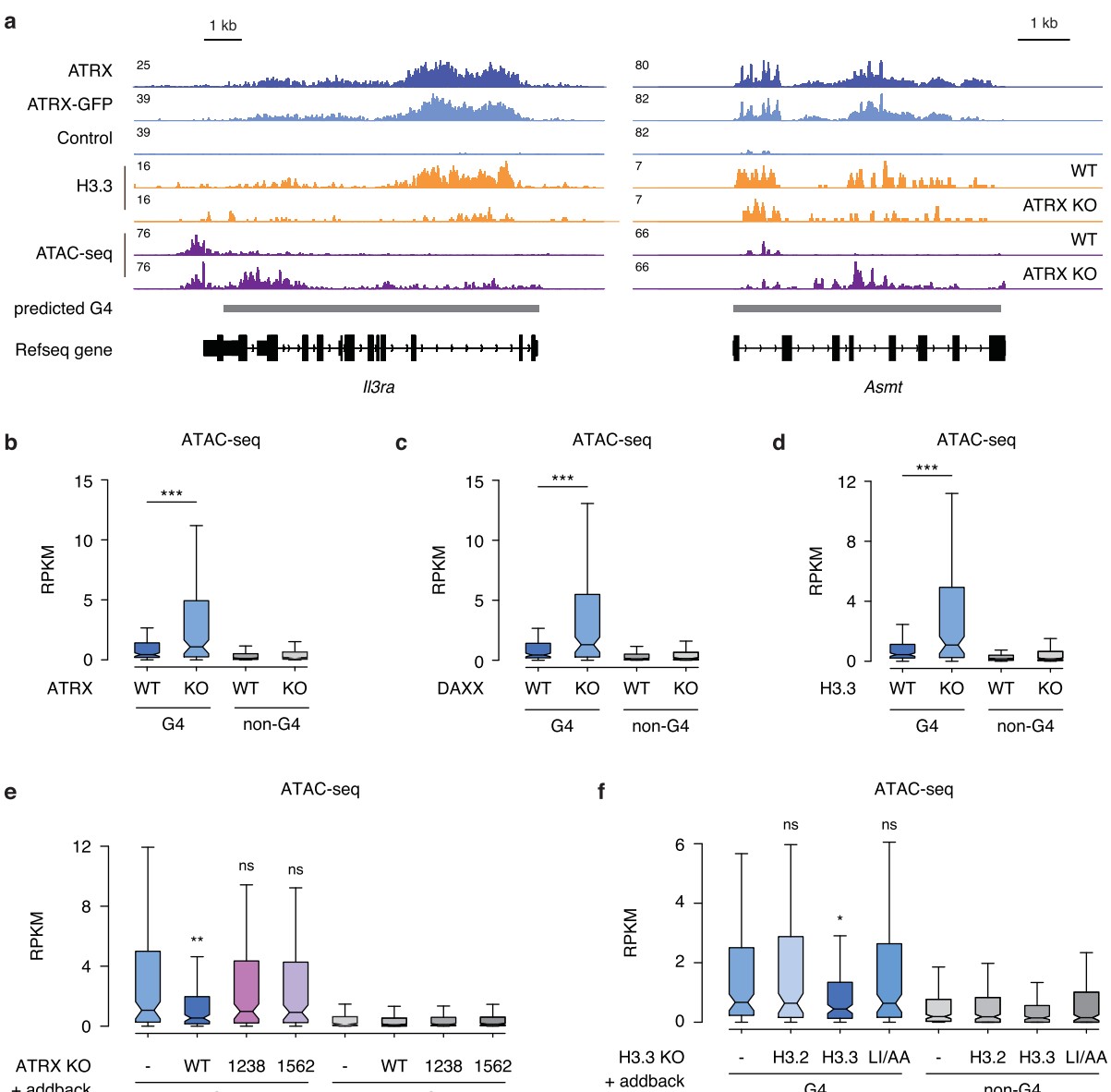

**Fig. 5 ATRX maintains closed chromatin states at G4 structures. a** Genome browser representations of ATRX, ATRX-GFP, H3.3 ChIP-seq and ATAC-seq at predicted G4 regions in ESCs. Box plots representing ATAC-seq read counts at ATRX-enriched G4 and non-G4 regions in wild-type ESCs compared to (**b**) ATRX KO ($p = 5.6 \times 10^{-4}$), (**c**) DAXX KO ($p = 1.7 \times 10^{-4}$), and (**d**) H3.3 KO ($p = 4.4 \times 10^{-5}$) ESCs. **e** Box plots representing ATAC-seq read counts at ATRX-enriched G4 and non-G4 regions in ATRX KO ESCs exogenously expressing either wild-type ($p = 0.003$) or mutant ATRX (L1238A $p = 0.47$; K1562R $p = 0.57$) constructs. **f** Box plots representing ATAC-seq read counts at ATRX-enriched G4 and non-G4 regions in H3.3 KO ESCs exogenously expressing either wild-type H3.2 ($p = 0.68$), H3.3 ($p = 0.047$), or mutant ($p = 0.87$) H3.3 constructs. Data are representative of two independent experiments. The bottom and the top of the boxes correspond to the 25th and 75th percentiles, and the internal band is the 50th percentile (median). The plot whiskers correspond to 1.5 interquartile range. Statistical significance determined by Wilcoxon–Mann–Whitney test compared to ATRX KO (**e**) and H3.3 KO in (**f**). *$p < 0.05$; **$p < 0.01$; ***$p < 0.001$. ns not significant.

regions. These demonstrate that deposition of H3.3 plays a role in maintaining a closed chromatin state in these regions. This is a somewhat surprising observation given that many regions of H3.3 deposition experience high rates of nucleosome turnover[56–58], and that in general, H3.3 is associated with open chromatin states[59,60]. These data do, however, support that keeping G4 regions heterochromatinized is likely a defining functional role of ATRX. Given the published literature[3,12,47], and our results in this study, it is likely that ATRX protects cells from replicative stress by doing so. Additionally, we observe increased chromatin accessibility and decreased H3K9me3 heterochromatin at ATRX-enriched G4 regions in ATRX helicase mutant-expressing cells. Our data show an increase in G4 at

EdU-synthesized DNA and G4-mediated replicative stress in helicase-mutant-expressing cells, as well. Based on these observations, it is likely that ATRX unwinds G4 in cells. This is in contrast to a previous study showing that ATRX is not able to unwind G4 structures in vitro[14]. This could be due to lack of nearby chromatin on the G4 DNA in vitro or perhaps because ATRX has a preference for a specific fold of G4 structure. Additional experiments are needed to test whether chromatinized DNA or other varieties of G4 DNA structures are requirements for ATRX helicase activity.

It is widely reported that ATRX mutation is strongly linked to human ALT cancers such as pancreatic neuroendocrine tumors and glioma[39]. Many works support the hypothesis that

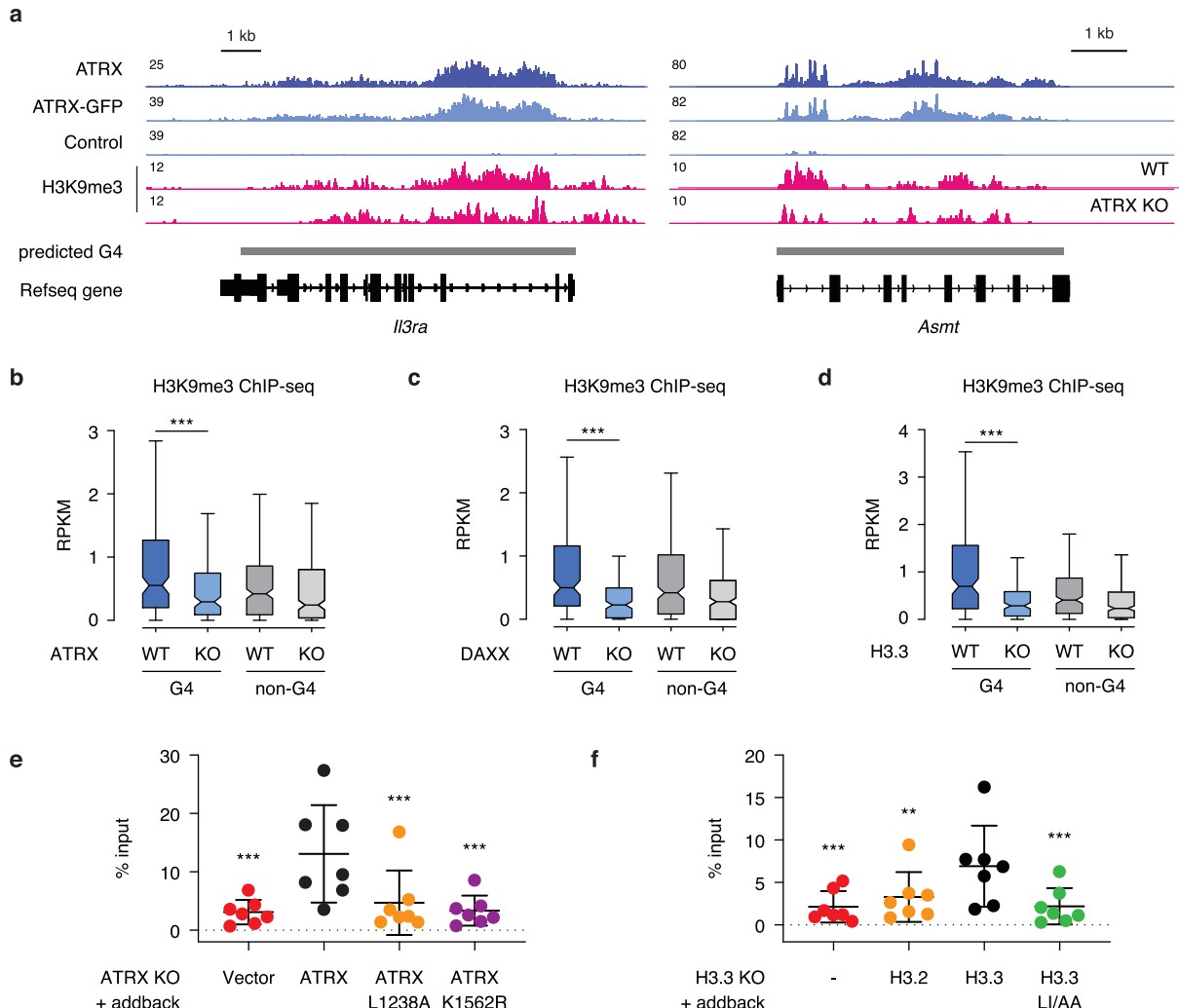

**Fig. 6 ATRX/DAXX/H3.3 are key factors for H3K9me3 maintenance at G4 regions. a** Genome browser representations of ATRX, ATRX-GFP, and H3K9me3 ChIP-seq at predicted G4 regions in ESCs. Box plots representing ChIP-seq read counts for H3K9me3[11] at ATRX-enriched G4 and non-G4 regions in wild-type ESCs compared to (**b**) ATRX KO ($p = 4.8 \times 10^{-4}$), (**c**) DAXX KO ($p = 3.1 \times 10^{-7}$) and (**d**) H3.3 KO ($p = 2.1 \times 10^{-8}$) cells. Data are representative of two independent experiments. The bottom and the top of the boxes correspond to the 25 and 75th percentiles, and the internal band is the 50th percentile (median). The plot whiskers correspond to 1.5 interquartile range. Statistical significance determined by Wilcoxon–Mann–Whitney test. ***$p < 0.001$. **e** ChIP-qPCR of H3K9me3 at ATRX-enriched G4 regions ($n = 7$) in ATRX KO ESCs exogenously expressing either wild-type or mutant ATRX constructs. **f** ChIP-qPCR of H3K9me3 at ATRX-enriched G4 regions ($n = 7$) in H3.3 KO ESCs exogenously expressing either wild-type H3.2, H3.3, or mutant H3.3 constructs. Data represent mean ± SD. Statistical significance determined by one-way ANOVA compared to ATRX KO + ATRX in (**e**) and H3.3 KO + H3.3 in (**f**). **$p < 0.01$; ***$p < 0.001$. $p = 2.9 \times 10^{-4}$ (vector vs. ATRX addback in **e**), $p = 1.8 \times 10^{-3}$ (ATRX addback vs. L1238A in **e**), and $p = 3.8 \times 10^{-4}$ (ATRX addback vs. K1562R in **e**). $p = 4.9 \times 10^{-4}$ (H3.3 KO vs. H3.3 addback in **f**), $p = 6.6 \times 10^{-3}$ (H3.2 addback vs. H3.3 addback in **f**), and $p = 5.6 \times 10^{-4}$ (H3.3 LIAA vs. H3.3 addback in **f**).

ATRX-deficient tumors show genome instability due dysregulation at specific repetitive DNA regions (e.g., telomere or ribosomal DNA)[39]. Our analysis of mutation at observed G4 regions in ATRX-deficient tumors are the first to show the direct link between ATRX mutation and G4 in tumor samples by whole genome analysis, with C/G to A/T mutation being common in ATRX-deficient tumors. This result is phenocopied in tumors containing mutation in ESET, supporting our model that heterochromatinization protects G4 DNA from replicative stress. In the future, it will be interesting to determine the molecular mechanism by which ATRX loss promotes substitutions at G4 regions through extensive biochemical and molecular biology approaches including careful analysis of the replication machinery and whole genome analysis in tumors.

Overall, we conclude that ATRX/DAXX protects G4 regions by maintaining a heterochromatin state marked by both H3.3

deposition and reduced chromatin accessibility. These data provide new insights into molecular mechanisms by which this complex supports genome integrity with important implications for our understanding of human disease.

## Methods

**Plasmids**. To tag the endogenous ATRX locus with GFP, pCAS9-mCherry empty, pCAS9-mCherry-Frame +1, and pCRISPaint-TagGFP2-PuroR plasmids were purchased from Addgene. To generate HA-tagged ATRX expression plasmid (PB-ATRX-HA-Neo), human ATRX cDNA were amplified from HeLa and assembled into PB-EF1α-MCS-IRES-Neo vector (System Biosciences) by Gibson assembly. The ATRX sequence corresponded to isoform 2 (NM_138270.4) and was verified by standard Sanger sequencing. Site-directed mutagenesis was used to generate mutations in PB-ATRX-HA-Neo. The ATRX plasmids were propagated using ElectroMAX™ Stbl4™ Competent cells (Life Technologies). To generate Flag-tagged DAXX expression plasmid (pCDH-Flag-DAXX-Puro), Flag-tagged DAXX DNA were amplified from plasmid Flag-Daxx/pRK5 (Addgene) and assembled into pCDH-EF1α-MCS-IRES-Puro vector (System Biosciences).

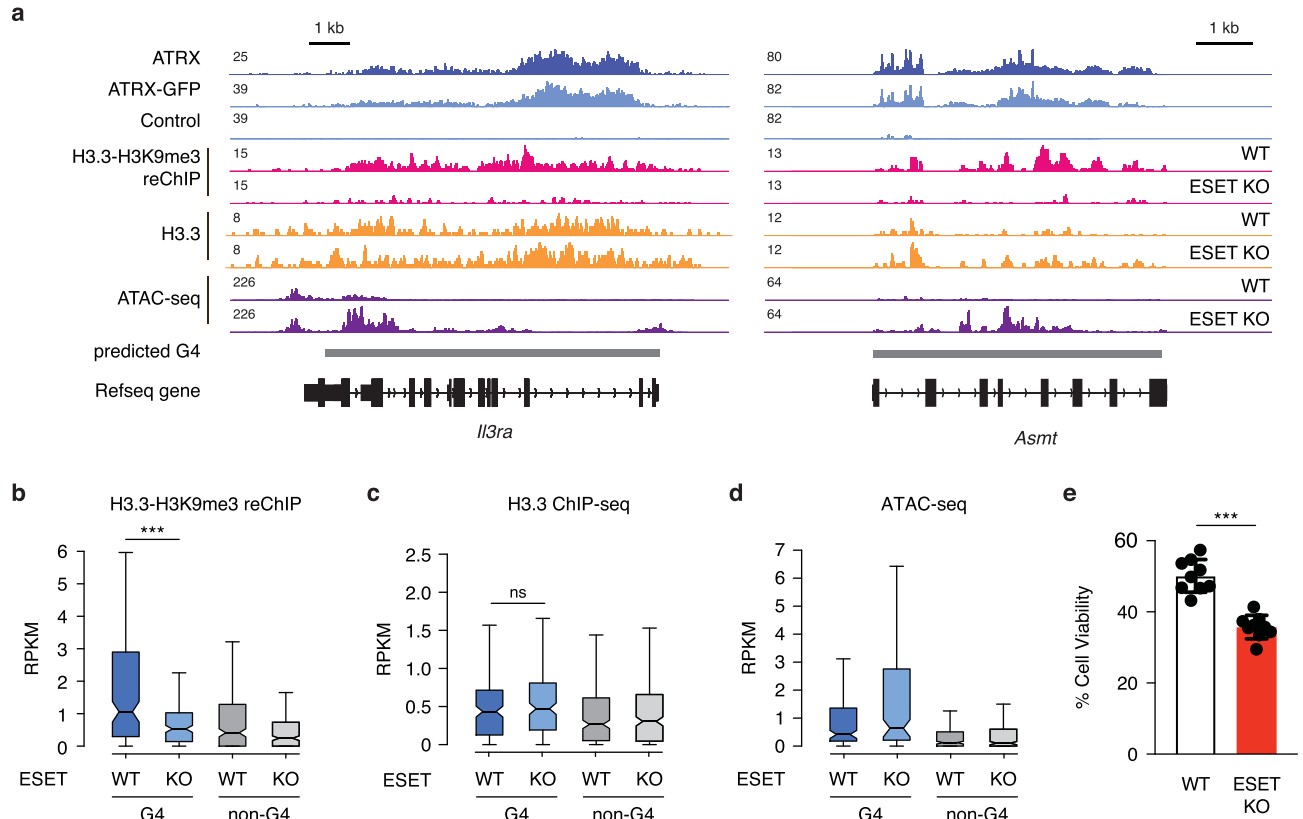

**Fig. 7 ESET facilitates heterochromatin at G4 regions and protects cells from G4-mediated stress. a** Genome browser representations of ATRX, ATRX-GFP, H3.3-H3K9me3 reChIP-seq, H3.3 ChIP-seq, and ATAC-seq at predicted G4 regions in ESCs. Box plots representing (**b**) H3.3-H3K9me3 reChIP-seq[11] ($p = 5.0 \times 10^{-6}$), (**c**) H3.3 ChIP-seq[11] ($p = 0.47$), and (**d**) ATAC-seq ($p = 0.12$) at ATRX-enriched G4 and non-G4 regions in wild-type ESCs compared to ESET KO ESCs. The bottom and the top of the boxes correspond to the 25 and 75th percentiles, and the internal band is the 50th percentile (median). The plot whiskers correspond to 1.5 interquartile range. Statistical significance determined by Wilcoxon–Mann–Whitney test. ***$p < 0.001$; ns: not significant. **e** Cell viability of wild-type and ESET KO ESCs treated with 1 μM of PDS for 3 days. Mock-treated cells at day 0 were taken as 100% survival. Data represented as mean ± SD ($n = 9$). Statistical significance determined by Mann–Whitney U test. ***$p < 0.001$. $p = 9 \times 10^{-10}$ (WT vs. ESET KO in **e**).

**Cell culture**. ESCs were cultured on gelatin-coated plates under standard conditions (KO-DMEM, 2 mM Glutamax, 15% ES grade fetal bovine serum, 0.1 mM 2-mercaptoethanol, 1× Pen/Strep, 1× NEAA and leukemia inhibitory factor) at 37 °C with 5% $CO_2$. ATRX KO, DAXX KO, HIRA KO (HIRA KO2), H3.3 KO, and ESET cKO cells have been described previously[3,48,61,62]. ESET deletion was induced by 1 μM 4-OHT (Sigma) treatment in ESET cKO cells. HeLa cells were purchased from ATCC and cultured in medium (DMEM, 10% fetal bovine serum, 1× Pen/Strep and 2 mM Glutamax) at 37 °C with 5% $CO_2$. Cells were tested routinely for mycoplasma.

**Stable cell lines**

*ATRX-GFP ESCs*. GFP-tagged ATRX ESCs were generated using CRISPR/Cas9-mediated homology directed repair[27]. $1 \times 10^6$ ESCs were mixed with 2.5 μg target selector plasmid (pCAS9-mCherry harboring ATRX guide RNAs (AAATCA-CATTGATTTCCCTG)), 2.5 μg frame selector plasmid (pCAS9-mCherry-Frame +1) and 5 μg donor plasmid (pCRISPaint-TagGFP2-PuroR). Electroporation was performed using the Neon transfection system (Thermo Fisher) with the settings: 1400 V, 10 ms pulse width, and 3 pulses. Cells were selected with 1 μg ml⁻¹ puromycin for 3 days. Single-cell clones were analyzed by genotyping to confirm GFP in-frame tagging of the ATRX locus and cell lysates of single clones were subjected to immunoprecipitation to confirm that GFP-tagged ATRX interacts with its complex partner, DAXX.

*Addback ESCs*. To generate exogenous wild-type or mutants HA-tagged ATRX in ATRX KO ESCs, $1 \times 10^5$ ESCs were co-transfected with 1.5 μg wild-type or mutants of PB-ATRX-HA-Neo and 1.5 μg Super PiggyBac transposase using Lipofectamine 3000 (Thermo Fisher). After 2 days of transfection, cells were selected with 300 μg ml⁻¹ Geneticin for 5 days and subjected to downstream analysis. Generation of stable H3.2, H3.3, and H3.3 mutant (L126A I130A, LI > AA) addback cell lines was described previously[10]. To generate exogenous DAXX in DAXX KO ESCs, cells were transduced with lentivirus encoding Flag-tagged DAXX and 8 μg ml⁻¹ polybrene overnight followed by selection with 1 μg ml⁻¹ puromycin for 3 days. To generate lentivirus, pCDH-Flag-DAXX-Puro plasmid (5 μg) and lentiviral components (5 μg psPAX2 and 1 μg VSVG) were transfected into $2 \times 10^6$ 293T cells in a 10 cm² dish using Lipofectamine 3000. Lentivirus-

containing supernatants from 48 and 72 h post-transfection were concentrated with Lenti-X (Clontech).

*ATRX KO HeLa cells*. To generate ATRX KO HeLa cells, $1 \times 10^5$ cells were transfected with SpCas9-2A-EGFP plasmid containing a single guide (CAGGATCGTCACGAT-CAAAGAGG) targeting exon 4 of the human ATRX gene using Lipofectamine 3000. Two days post-transfection, single cells were sorted onto gelatin-coated 96-well plates using a FACSAria Fusion cell sorter according to the instruction of the UT Southwestern Flow Cytometry Facility. Clones were expanded, genotyped, assessed for protein levels, and subject to downstream analysis.

**Co-immunoprecipitation**. Soluble nuclear proteins were extracted using ammonium sulfate[11]. For IP-MS, 5 mg of pre-cleared HeLa nuclear extracts were incubated overnight at 4 °C with either 40 μg ATRX antibody or rabbit IgG cross-linked to Dynabeads Protein G (Invitrogen). Beads were washed three times with 5 ml wash buffer (20 mM HEPES, pH 7.9, 150 mM KCl, 0.01% NP-40, 10% glycerol, 1 mM DTT, 0.4 mM PMSF) and once with TE buffer. Samples were eluted in 0.1 M glycine, pH 2 and neutralized with 1 M Tris, pH 8. To verify ATRX binding specificity, 10% of immunoprecipitated samples were subjected to immunoblot assay. Remaining samples were run resolved by SDS-PAGE and stained with Coomassie Blue R-250. The desired bands were excised and subjected to Thermo Orbitrap Fusion Lumos mass spectrometer. Protein identification was based on counts and sequences of peptides according to pipelines of UT Southwestern Proteomics Core Facility. For standard immunoprecipitation, 0.5 mg of nuclear extracts were incubated with 2 μg antibody bound to 10 μl Dynabeads Protein A or Protein G at 4 °C for 2 h. Beads were washed three times with 1 ml wash buffer and eluted in 0.1 M glycine, pH 2. Samples were neutralized with 1 M Tris, pH 8, denatured in SDS loading buffer and subjected to immunoblot assay.

**Mass spectrometry analysis**. Raw mass spectrometry data files were converted to a peak list format and analyzed using the central proteomics facilities pipeline, version 2.2.0. Peptide identification was performed using the X!Tandem and open MS search algorithm search engines against the human protein database from Uniprot, with

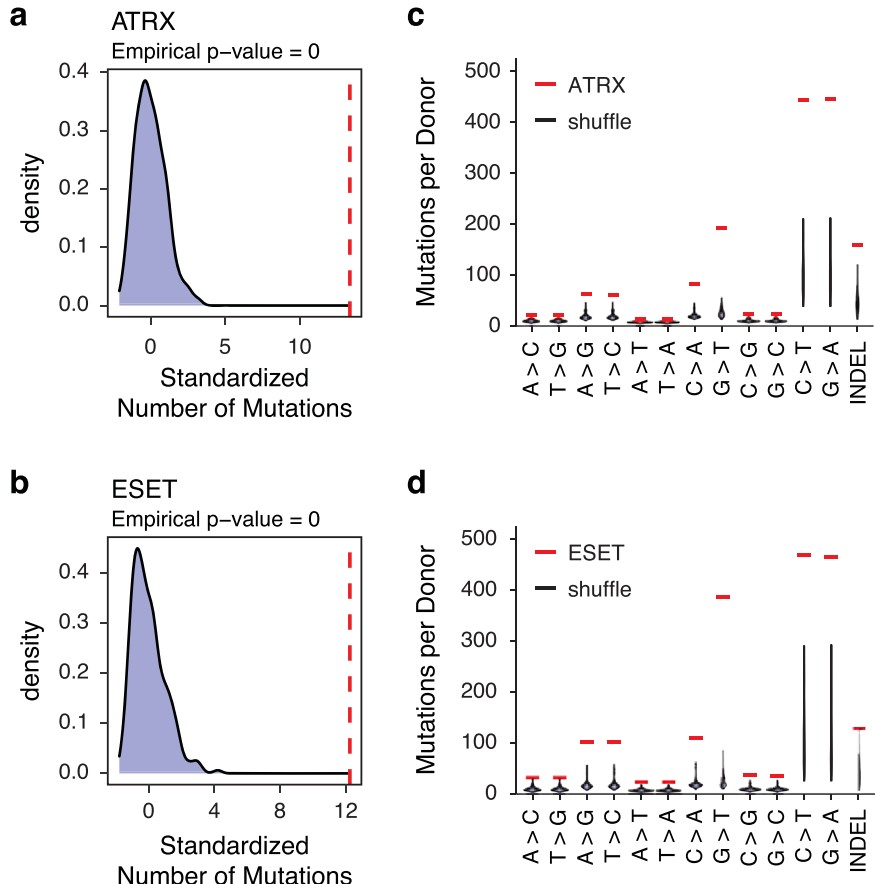

**Fig. 8 Mutations at G4 regions are highly correlated with mutations in *ATRX* or *ESET* in human tumors. a, b** Red line shows standardized number of mutations at G4 regions in *ATRX* ($n = 684$) (**a**) or *ESET* ($n = 312$) (**b**) mutant tumors. Histograms show the mutation density at observed G4 regions in an iteratively ($n = 504$) and randomly selected patient cohort of the same size. *p* value calculated based on the null distribution as described in methods. **c, d** Analysis of single-nucleotide mutations and insertion-deletion (INDEL) mutations in the *ATRX* (**c**) or *ESET* (**d**) mutant tumors compared with the shuffled patient cohort described above. Mutations per donor from the shuffled cohort are displayed as a violin plot in which the width of the shaded area represents the proportion of data located there.

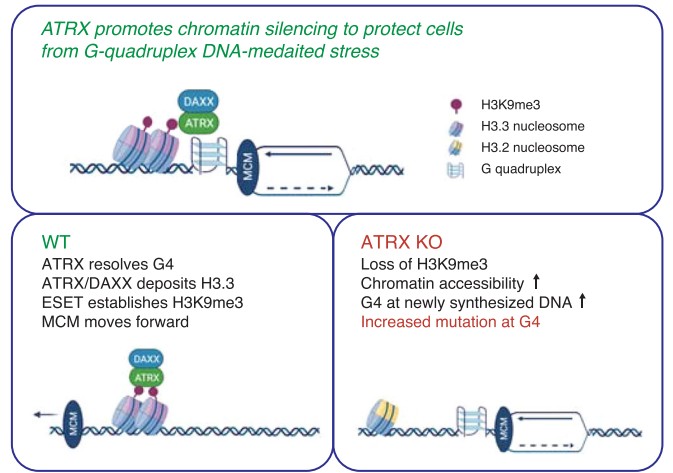

**Fig. 9 ATRX is a critical chromatin remodeler at G-quadruplex regions during DNA replication.** When the replicative MCM helicase complex encounters G4 DNA, ATRX resolves G4 DNA through its helicase activity and H3.3 deposition activity to facilitate MCM progression. These activities are ultimately upstream of ESET-mediated heterochromatin formation to protect cells from G4-mediated replicative stress. Thus, ATRX/DAXX/H3.3 and ESET cooperate to prevent physical G-quadruplex stress and maintain genome stability.

stable contaminants and reversed decoy sequences appended. Fragment and precursor tolerances of 20 ppm and 0.5 Da were specified, and three missed cleavages were allowed. Carbamidomethylation of Cys was set as a fixed modification and oxidation of Met was set as a variable modification. Label-free quantitation of proteins across samples was performed using SINQ normalized spectral index software.

**Proximity ligation assay (PLA).** Cells were seeded on glass coverslips coated with 8 µg ml$^{-1}$ fibronectin in 12-well plates. Cell cycle synchronization was performed by incubating cells with 2 mM thymidine for 14 h and following 50 ng ml$^{-1}$ nocodazole treatment for 7 h[63]. Mitotic cells were released and newly synthesized DNA was labeled with 2 µM EdU for 20 min in prior to fixation. Cells were fixed with 4% paraformaldehyde in PBS pH 7.4 for 10 min at room temperature then permeabilized in 0.5% Triton X-100 in PBS for 15 min.

For ATRX and G4 foci, cells were fixed and permeabilized as described previously. To detect newly synthesized DNA, EdU was clicked with 488-azide using a Click-iT™ Cell Reaction kit (Thermo Fisher, C10269) according to the manufacturer's protocols. Cells were incubated with 100 µg ml$^{-1}$ RNase A (Sigma, R4642) for 1 h at 37 °C and then washed in PBS four times for 5 min each. Cells were blocked in Duolink blocking solution for 1 h at 37 °C and then incubated with 0.06 µg BG4 antibody for 1 h at 37 °C, then with mouse anti-ATRX antibody diluted 1:100 and rabbit anti-DYKDDDDK antibody diluted 1:1000 overnight at 4 °C. All primary antibodies were diluted with Duolink Antibody Diluent. PLA foci were developed with Duolink In Situ Red Starter Kit (Sigma, DUO92101) according to the manufacturer's protocols. The EdU and PLA fluorescence signals in representative images for Fig. 1e, Fig. 3a, Supplementary Figs. 2c, 7a were pseudocolored red and green, respectively, to clearly show PLA foci.

For detecting G4 at newly synthesized DNA, in situ analysis of protein interactions at DNA replication forks (SIRF) was used[43]. Cells were fixed and permeabilized as described previously. Biotin-azide added to the Click-iT reaction cocktail and incubated with cells. RNase A treatment, cell blocking and BG4 antibody incubation were the same as described previously. Cells were incubated with 1:1000 of mouse biotin antibody and 1:1,000 of rabbit anti-DYKDDDDK

antibody in Duolink Antibody Diluent overnight at 4 °C. For normalization of G4-EdU foci, cells were separately incubated with 1:1000 of mouse biotin antibody and 1:1000 of rabbit biotin antibody. The PLA foci were developed with Duolink In Situ Red Starter Kit. The cover glass was mounted with the mounting medium containing DAPI in the dark overnight and kept at 4 °C or −20 °C for imaging.

**Imaging and processing**. Cells were z-stack imaged using a DeltaVision Elite Deconvolution microscopy (GE Healthcare). Images were further processed by deconvolution using softWoRx software. Mean fluorescence intensities (MFI) of PLA foci between G4 and EdU were counted using maximum intensity z-projection of ImageJ. At least 30 nuclei were counted for each condition. Then the data were normalized to MFI of PLA foci between EdU and EdU in which we used mouse and rabbit antibodies against biotin-clicked EdU.

**Cell viability assay**. $0.5 \times 10^3$ cells were treated with serial dilutions of PDS (Sigma, SML0678) for 5 days on a 96-well plate. Cell viability was analyzed by MTT assay (Sigma, M2128).

**Chromatin immunoprecipitation (ChIP)**. Crosslinking ChIP was performed according to published methods[2,64]. Native ChIP was performed according to published methods[11]. Details are described in Supplementary Methods.

**ChIP-seq and data analysis**. ChIP-seq libraries were prepared from 5 ng ChIP DNA following the Illumina TruSeq protocol. The size of libraries was determined using a D5000 ScreenTape on a 2200 TapeStation (Agilent) and the amount of libraries was quantified using a Qubit dsDNA HS Assay kit (Thermo Fisher). Libraries were paired-end 33-base sequenced on the Illumina NextSeq 500. Typical sequencing depth was at least 20 million reads per sample.

*ChIP-seq data quality control, alignment, and spike-in normalization*. Quality of ChIP-seq data sets was assessed using the FastQC tool (v.0.11.2). ChIP-seq raw reads were aligned separately to the mouse reference genome (mm10) and the spike-in Drosophila reference genome (dm3) using BOWTIE2 (v.2.2.8). Only one alignment is reported for each read (either the single best alignment or, if more than one equivalent best alignment was found, one of those matches selected randomly). Duplicate reads were filtered using the MarkDuplicates tool of Picard (v.1.127). Uniquely mapped Drosophila reads were counted in the sample containing the least Drosophila mapped reads and used to generate a normalization factor for random downsampling. Reads were converted into bedgraph files using BEDTools (v.2.29.0) and then converted to bigwig using bedGraphToBigWig utility of UCSC kent tools (v317) for visualization in Integrative Genomics Viewer (v.2.3).

*Peak calling*. ATRX ChIP and ATRX-GFP ChIP samples were merged using MergeSamFiles tool of Picard (v.2.10.3). Peak calling was performed on the merged files using MACS software (v.1.4.2) using cutoff values "—p value 1e−5—mfold 10, 30" and GFP ChIP in ESCs (without GFP) as input.

*Box plots*. Box plot representations were used to quantitatively assess the read distribution. Box plots are defined by the median, box limits at upper and lower quartiles of 75 and 25%, and whiskers at 1.5× interquartile range. The read distribution of the peaks was calculated and plotted using custom R scripts.

*Average profiles*. BigWig files were used to generate average ChIP-seq profiles using deepTools (v.3.3.0). The peaks were scaled to the average length of peaks (1300 bp). Read densities surrounding 4 kb (±2 kb) of the scaled peaks were determined and visualized.

*G-quadruplex prediction*. G4 motif was predicted by modifying the Quadparser available at: https://github.com/dariober/bioinformatics-cafe/tree/master/fastaRegexFinder. Sequences within peak regions were analyzed by regular expression matching for the G4 motif defined below. The peaks were then assigned to one of two groups: either a G4-containing peak (presence of at least one motif) or a non G4-containing peak (absence of a G4 motif). G4 motif is defined as: "([gG]{3,}\w{1,12}){3,}[gG]{3,}". The regex looks for three or more runs of guanines followed by 1–12 of any other bases. This is repeated three or more times, ending with three runs of guanines.

*G-quadruplex enrichment analysis*. Simulation was used to estimate if the number of G-quadruplex motif is considerably enriched in the ATRX peaks. The peaks were shuffled using the BEDTools shuffle command at random throughout the genome while maintaining the number of peaks and their size and the number of G-quadruplexes computed. The fold-enrichment analysis was calculated by comparing the actual count of G-quadruplex motif within ATRX peak regions to the average count of the same peak regions after hundred randomizations.

*Motif analysis*. Multiple EM for motif elicitation (MEME) was used to interrogate the peak sequence data set for recurring motifs across G4 and non-G4 peaks.

*Peak annotation*. Peaks were annotated to nearest genes using HOMER annotatePeaks.pl with default settings.

**ATAC-seq and analysis**. ATAC-seq was performed and analyzed as described in Supplementary Materials[65].

**CUT&Tag and analysis**. CUT&Tag was performed and analyzed as described in Supplementary Materials[66,67].

**Analysis of mutation rates from patient data**. Observed G-quadruplex regions in the human genome were downloaded from GSE110582 in the form of bedfiles. G4 regions enriched after PDS treatment were identified using G4-seq2 methodology[68]. For each gene, donors were selected from the International Cancer Genome Consortium Data Portal[69] who were classified as having high impact functional mutations in that gene. All simple somatic mutations (SSMs) within that donor set were then intersected with the above G4 regions using bedtools intersect −u. The total number of substitutions and INDELs for each gene's donor set were counted. Likewise, for each gene, null donor sets were created by randomly sampling the same number of donors from the total pool of 19,729 ICGC donors with available SSM data and counting G4-intersecting substitutions and INDELs within that set. This was repeated 504 times for each gene in order to create a null distribution for that gene.

Null distribution for each gene was standardized by centering at the mean and scaling by standard deviation (s.d.). For the test observation itself (red line), we standardized by subtracting the mean of the null and scaling by the s.d. of the null. We calculated the $p$ value from first definitions by calculating the area under the curve to the right of the observed point using the null distribution.

**Statistical analysis**. Data analysis was performed using Prism 8 for PLA, MTT assay and ChIP-qPCR. Quantification of PLA data is presented as scatter dots on box-and-whisker plots marking a horizontal median line. In box and whisker plots, box and whiskers indicate 25–75 and the minimum and maximum values, respectively. Statistical significance is determined by One-way ANOVA test or two-tailed Mann–Whitney $U$ test, where noted. Number of individual values for each experiment are reported in the figure legends. For MTT data, mock-treated cells at day 0 were taken as 100% survival. Quantification of MTT data is presented as a XY graph or column graph with mean ± SD. Statistical significance is determined by Two-way ANOVA test. Data are representative of three independent experiments unless indicated in the figure legends. Half-maximal inhibitory concentration (IC50) was determined by nonlinear regression. Statistical significance is determined by One-way ANOVA test for ChIP-qPCR. Wilcoxon rank sum test was used to calculate p values for all comparisons in NGS data sets. $*p < 0.05$; $**p < 0.01$; $***p < 0.001$; ns not significant.

**Data sets**. The following published next-generation sequencing data sets were meta-analysed in this study: (1) ChIP-ATRX in ESCs[2], H3K9me3 for WT, ATRX KO, DAXX KO, H3.3 KO, ESET KO[70], and ESET KO reChIP[11], H3.3 for WT, ATRX KO, DAXX KO, ESET KO[11,61] and H3K27ac for WT, ATRX KO and DAXX KO[61], (2) RNA-seq in WT, ATRX KO, DAXX KO and H3.3 KO[61,71], (3) ATAC-seq in HIRA KO1, H3.3 KO[61], and H3.3 addback[10], (4) SNS-seq in ESCs[45], (5) G4-seq in human cells[68], and (6) Pan-cancer analysis of whole genome[72]. Also, Supplementary Table 1.

**Reporting summary**. Further information on research design is available in the Nature Research Reporting Summary linked to this article.

## Data availability
Data files have been deposited in the NCBI Gene Expression Omnibus database under series accession number GSE151058. Subseries are GSE151053 (ATAC-seq), GSE151054 (ChIP-seq), GSE151056 (EdU-seq), GSE171461 (CUT&Tag). The mass spectrometry proteomics data have been deposited to MassIVE with the data set identifier MSV000085543. Source data are provided with this paper.

## Code availability
Code to generate figures is available at https://github.com/utsw-medical-center-banaszynski-lab/Teng-et-al-2021-Nature-Communications

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

## Acknowledgements

We thank members of the L.A.B. lab and H. Yu for helpful discussions; E. Duncan for critical comments on the paper; UTSW BioHPC for computational infrastructure; UTSW McDermott Center for providing next-generation sequencing services; UTSW Flow Cytometry Core; UTSW Live Cell Imaging Core; UTSW Proteomics Core for mass spectrometry. Figure 1d, Supplementary Fig. 2a, and Fig. 9 were created with BioRender.com.

L.A.B. is a Virginia Murchison Linthicum Scholar in Medical Research (UTSW Endowed Scholars Program), an American Cancer Society Research Scholar, and a Peterson Investigator of the Neuroendocrine Research Foundation (NETRF). This work was supported in part by CPRIT RR140042, The Welch Foundation I-1892 and I-2025, and NIH R35 GM124958 (L.A.B.), the Taiwan Postdoctoral Research Abroad Fellowship (Y.-C.T.), the American-Italian Cancer Foundation and the Center for Regenerative Sciences and Medicine at UTSW (S.M.), and the Green Center for Reproductive Biology Sciences.

## Author contributions

Y.-C.T. and L.B. conceived and designed the study; Y.-C.T. performed experiments with help from V.G., M.L., S.M., and J.W.; A.S. and R.O. performed computational analysis; R.O. and A.B. performed patient mutation analysis; L.A.B. supervised and provided funding for the project; Y.-C.T and L.A.B. wrote the paper with input from all authors.

## Competing interests

The authors declare no competing interests.
