## [Peer Review File · Nature Communications]

REVIEWER COMMENTS

Reviewer #1 (Remarks to the Author):

Several biological functions of ATRX have been described in detail. These include ATRX as a chromatin remodeller, as histone chaperone involved in the deposition of histone H3.3, it localizes to heterochromatin and acts as a helicase resolving G-quadruplex (G4) structures during replication. Mutations in ATRX leads to severe human diseases. The authors of this manuscript address questions related to these described functions of ATRX with a special focus on the relation between ATRX and G4 structures. They end up with a very nice model figure on the function of ATRX, but on the way to this model several problems of this manuscript have to be clarified. These include

Page 3/4 the data clearly show that ATRX colocalizes with G-rich and potential G4 forming sequences. But they do not tell whether ATRX indeed binds to G4 structures. Why are no ATRX/G4 stainings shown for G1 and G2 phase in Figure 1d. It would be expected that the number of G4 spots decrease during S phase but this is not clear from the figure. A statistical analysis the number of ATRX/G4 spots during all phases of the cell cycle would be needed. How does the staining look like in metaphase chromosomes? Finally, before asking whether ATRX binds to G4 in vivo authors should show that it binds to various G4 structures in vitro.

Page 4: Minor: what means „replicative role“?

Page 3-5: ATRX seems to interact with a large number of other proteins and most of these interactions are required for proper ATRX function. Have the authors made any attempt to identify which region of ATRX interacts with which other protein. And how are the interactions between ATRX interaction partners. These questions may not be essential for aims of this manuscript, but they should be at least mentioned.

Page 5/6: from the data presented it is not clear whether there is a direct interaction of ATRX with the MCM helicase or whether this interaction occurs via DAXX. Biochemical analyses would be required in addition to the presented immunoprecipitation experiments. In addition, which helicase (ATRX or MCM) resolves G4 structure as claimed in Figure 4 (see also general comment).

Page 6: Authors claim that G4 structures accumulate at replication sites in the absence of ATRX. They conclude that the ATRX/DAXX complex is required to remove (resolve) G4 structures during early replication. This is supported by one microscopic figure. This is not acceptable for such a claim. Either many nuclei from ATRXO or DAXXO cells have to be shown or a panel of such nuclei should be presented. The G4 spots probably should represent G4 structures in replication factories. However, when looking at replication factories in the literature they are much smaller and more distinct. Authors should show G4 signals in normal replicating cells and subsequently they have to explain the strange looking G4 signals in their knock out cells.

Pages 6-9: Figure 4 claims that ATRX/DAXX is required for removing G4 structure at sites of replication. In the following experiments evidence is provided that ATRX is involved in the deposition of H3.3 and maintenance/establishment of a heterochromatin at these sites. Authors mention that G4 structures form in nucleosome-free chromatin and that the establishment of heterochromatin may remove G4 structures. The model implies that ATRX is recruited to sites of some G4 structures and these are removed by the deposition of H3.3.

General comment: this manuscript contains some severe inconsistencies which have not only explained but also experimentally addressed. Figure 4 shows that G4 structures accumulate in the absence of ATRX suggesting an active role of ATRX (or the MCM helicase) in resolving this structure. G4 preferentially forms in nucleosome free regions of the genome. But it is explained that ATRX maintains a heterochromatic structure at G4 sites. Therefore it could be that the effect described in Figure 4 is that since heterochromatin is no longer retained in the absence of ATRX G4 structure could now become formed. Or does the formation of heterochromatin by ATRX resolves

G4 structures – a very novel mechanism. Experimental answers to these and the other question mentioned above have to be incorporated into a revised version.

In addition, most of the claims come from immunoprecipitation experiments or not really convincing immunofluorescence pictures. Additional biochemical methods should be used to support the claims.

In summary, this manuscript shows some nice observations but as it is in its present form it gives no more mechanistic explanations about the function of ATRX as already present in the literature.

Reviewer #2 (Remarks to the Author):

The ATRX-DAXX complex assembles the histone H3.3 variant into nucleosomes in heterochromatin regions of the genome. Loss of function mutations in ATRX and DAXX support a role for this complex in suppression of tumourigenesis. However, their functional role as possible tumour suppressors remains largely uncharacterised. ATRX contains ATPase activity of unknown function. Previous studies have found that ATRX deficiency sensitizes cells to small molecules that bind to and stabilize G-quadruplex structures, suggesting that ATRX may resolve these structures. However, the previous data showing interaction between ATRX and G-quadruplexes is generally of low quality, and whether ATRX directly interacts with these structures remains debated. Previous studies have also linked loss of ATRX to genomic instability, possibly caused by defects in DNA replication fork progression.

Here, Tang and colleagues explore the role of ATRX and DAXX in G-quadruplexes in mouse ES cells. The authors found that 40% of ATRX ChIP seq peaks are near sequences that have the propensity to form G-quadruplexes. The majority of ATRX peaks don't overlap with predicted G-quad structures, but rather with what appears to be simple repeats. The authors found that loss of ATRX causes subtle reductions in H3K9me3 and H3.3 ChIP-seq signal at many putative G-quad sequences. ATRX transgenes that interfere with DAXX interaction and ATPase activity exhibit similar sensitivity as the ATRX knockout cells to a G-quadruplex stabilizing molecule (pyridostatin), suggesting that ATRX may resolve these structures. Using co-IP experiments followed by mass spectrometry analysis to determine associating proteins, the authors found MCM proteins and the two subunits of the FACT histone chaperone complex. Knockdown of FACT subunits didn't affect ATRX-DAXX association with G4 structures by ChIP-seq, and the functional significance of ATRX-FACT interaction remains unknown. Using SNS-seq, the authors found DNA replication origins near some ATRX-binding sites. EdU labeling and sequencing analysis suggests that some origins may initiate replication earlier in the absence of ATRX.

The strength of this study is the use of high resolution ChIP seq to map ATRX-DAXX and H3.3 to regions of the genome that likely have the propensity to form G-quadruplexes. Unfortunately, the authors don't use the G-quadruplex antibody for ChIP-seq in their knockout ATRX, DAXX, and H3.3 cell lines to specifically address if loss of these proteins leads to increased G-quadruplex structures across the genome. Instead, they use the G-quad antibody in relatively low resolution proximity ligation assay in order to make conclusions about how ATRX-DAXX G-quadruplex structures and origin function (as measured by EdU incorporation). Previous studies have used the BG4 antibody to map G-quadruplex structures by next generation sequencing, and the authors need to determine if the loss of DAXX, ATRX, or ATRX ATPase activity affects the overall number of G-quadruplex structures. The PLA data is not convincing, and as I mention below various controls that are missing for these assays.

Numerous biochemical and genetic studies over the past twenty some years have demonstrated that the active MCM helicase involved in DNA replication origin firing consists of MCM2,3,4,5,6,7. However, only MCM2,4,6,7 are mentioned as interacting with ATRX. It is known that sub complexes of the MCM proteins exist, and some biochemically abundant sub complexes of unknown function (such as MCM2,4,6,7) can occur outside of the origin-competent full MCM hexamer consisting of MCM2-7. The authors finding (fig S4D) that ATRX does not interact with MCM3,5 really calls into question the model presented, namely that ATRX-DAXX interacts with the MCM proteins at origins of replication. Moreover, the finding of little EdU incorporation near sites of ATRX and MCM (Fig 1b) also suggests that ATRX is not present at active origins of replication. The

SNS-seq data do suggest that some origins are located near some ATRX binding sites, but additional analyses that quantitate these findings may help shed light on the functional significance of these findings.

Because the authors don't know how ATRX-DAXX interacts with the MCMs, experiments that could tease out the functional significance of this interaction remains unclear. As I wrote above, the evidence presented that ATRX-DAXX impacts origin function is currently weak. However, I think the manuscript could be greatly improved if the authors should pursue high resolution ChIP seq experiments to test their model the necessity of the ATRX-DAXX complex and ATPase activity for resolving G-quadruplex structures in vivo. Without additional experiments, the findings regarding ATRX-DAXX resolution of G-quadruplexes remains undeveloped and the study represents a very limited advance in our understanding of ATRX-DAXX complex function. These issues and other issues noted below should be resolved prior to consideration of acceptance of this manuscript.

Other comments and issues that should be resolved:

1) The authors found an enrichment of predicted G-quadruplex structures near 40% of ATRX ChIP seq peaks. The majority of ATRX peaks don't overlap with predicted G-quad structures, but rather with what appears to be simple repeats. What chromatin features are found at these simple repeats that seem to promote ATRX binding? Previous studies have found that the ATRX ADD domain has high affinity for H3K9me3, and the authors genome browser screenshots show H3K9me3 at predicted G4-quadruplex structures. Genome-wide, what is the overlap between H3K9me3 and the predicted G-quadruplex structures? Is H3K9me3, rather than G-quadruplex, a predictor of ATRX ChIP seq peaks? The authors should perform analyses comparing the ChIP seq data sets.

2) The authors model- that ATRX-DAXX interacts with MCMs at origins of replication also predicts that H3.3 should be enriched at these origins. Is H3.3 preferentially found at origins of replication? Do MCM proteins ChIP to G-quad or other genomic regions where ATRX ChIPs? For example, the genomic regions found in the SNS-seq analysis.

3) The authors use SNS-seq to identify origins of replication near ATRX-enriched G-quadruplexes. Figure 6d explores possible origins near ATRX-enriched G-quads, however, it's unclear how many 'ATRX-bound G-quad' genomic regions were used for this analysis. What proportion of total G-quad sequences are found in the SNS-seq data? This type of analysis would be important in order to determine putative origins are near G-quad sequences and ATRX binding sites.

4) The authors should provide screen shots of EdU seq and SNS seq at putative origins in WT and ATRX delete cells.

5) Previous studies have found that MCM2 has histone H3-binding activity, and it remains unclear from the experiments presented here that the MCM2,4,6,7 association with ATRX-DAXX isn't simply through bridging via interaction with H3.3-H4. The authors should determine if ATRX and DAXX can still associate with the MCM proteins in the absence of H3.3.

6) The PLA lacks important controls that are needed in order to assess the signal from these experiments. For example, the control assays should be conducted with only one of the antibodies. Additionally, the assay should be performed in ATRX deficient cells in order to control for possible ligation artifacts caused by non-specific antibody interactions. Based solely on the evidence provided in the manuscript, it's not clear that there is a substantial overlap between ATRX and EdU staining in late S phase. Thus, quantitation of this assay is required in order to determine the statistical significance of this result.

7) Figure 2A pyridostatin assay of ESCs needs to include IC50 values for the different genotypes.

Reviewer #3 (Remarks to the Author):

In this manuscript, Teng et al address ATRX function and mechanism at G4 structures. They show that ATRX is in close proximity to G4 structures in vivo and that this depends on the DAXX binding and helicase activity of ATRX. They identify MCM and FACT as ATRX/DAXX interactors and that ATRX functions to maintain G4 chromatin in a closed state and that loss of ATRX results in G4 accumulation at DNA replication forks. Some of the results presented are expected, others are intriguing. The authors make some overstatements and some of the experiments lack critical controls. However, with significant revisions, this manuscript can be considered for publication.

1. Fig 1 - The authors state that ATRX and G4 associate in all phases of the cell cycle. Can they show a cell cycle phase where there is not much interaction eg – G2/M, where G4 levels should be low due to lack of DNA replication? The authors have to show lack of association between another unrelated chromatin bound protein (HIRA?) and G4 to show that this association is specific to ATRX.
2. Fig 1- The treatment with DNaseI abolishes PLA signal. Does it abolish interaction signal between two proteins eg. ATRX and DAXX? Does DNase treatment abolish EdU signal (since this appears to be done only in G1 samples). This is required to show the specificity of DNase treatment to protein-G4 structures and not a non-specific effect on all PLA derived signal. All PLA images must be quantified (Figs. 1e, 3c, 3f to show number of EdU positive and PLA signal positive cells). Show different channels for IF. The merged images are not easy to assess.
3. Pg 4 – “Further, mutations to ATRX that disrupt its helicase activity (K1562R, K1612N) remain sensitive to PDS, in support of the hypothesis that ATRX unwinds G4 structures in vivo (Fig. 2b and Supplementary Fig. 3d)”. This is an overstatement. There is no evidence in this manuscript that ATRX is able to unwind G4 structures. It could inhibit G4 formation or recruit another helicase that does the unwinding. This conclusion must be rephrased.
4. Supplementary Fig 4d – show ATRX western. Also show DAXX western to show that ATRX can IP a known interactor when it cannot IP mcm3 and mcm5. Supplementary Fig 4e – show DAXX western to confirm that DAXX is being IPed. Also, as the authors stated, DAXX binds numerous proteins in the cell independent of ATRX. One of these should be used as a control in DAXX IPs.
5. Both MCM6 and FACT are abundant proteins. I am not sure that PLA with these proteins and another chromatin bound protein (ATRX) is even meaningful. Can the authors show another chromatin bound protein that does interact with FACT components, but interacts with its cognate interactor?
6. DAXX binding mutant L1238A and Helicase mutant K1562R, K1612N are sensitive to PDS. Does treatment of cells with PDS increase G4 to the same extent? Can the BG4 antibody be used for a dot plot analysis and levels of G4 quantified?
7. “Loss of ATRX leads to accumulation of G4 at replication sites” – This entire section would benefit from genomics or significantly higher resolution microscopy techniques. PLA experiments alone are not convincing. To state that G4 structures accumulate at replication sites, a detailed comparison of G4 ChIP (using BG4 or other) and some replication complex protein must be shown. Does MCM complex KD also result in G4 accumulation and/or ATRX recruitment as the model would suggest?
8. Pg 6 - ‘Our identification of an association between ATRX and the MCM and FACT complexes add a direct link to the existing literature suggesting that ATRX plays a role in DNA replication’. I think there is enough literature showing (and not merely suggesting) that ATRX loss does play a role in DNA replication.
9. Supplementary Fig 6 – Quality of PLA images are not convincing and need to be more clear.
10. Fig. 5 – The ATAC Seq data is very compelling here and clearly shows increased accessibility at sites that contain G4 and ATRX. Can this be rescued by overexpression of DAXX (or H3.3) that is known to deposit variant histones (CENPA) non-specifically when it is overexpressed.
11. The authors suggest that ATRX/DAXX does not contribute uniformly to transcriptional activity of genes containing ATRX enriched G4 regions. The authors should consider that their RNA seq looks at steady state levels and not rates of transcription. Since ATRX has been shown previously to facilitate elongation through structured DNA, it is possible that changes will only be visible using methods such as Gro-seq or Pro-seq. This may be worth discussing further.
12. Fig 6 – Can levels of G4 be tested by BG4 ChIP next to some of the replication origins in WT and ATRX KO? Do G4 levels go up?

Please find our revised manuscript for further editorial consideration. We were encouraged by reviewer comments, and overall feel that their suggestions have greatly improved the rigor of our work, the conclusions that we are able to draw, and the contribution of this work to the field.

Based on reviewer suggestions, we have substantially revised our manuscript, outlined in more detail below. This includes major experimental and computational revision. The main points of our revised submission are:

- 1) ATRX is bound at G4 regions
- 2) ATRX associates with the MCM helicase
- 3) Loss of ATRX results in increased association between G4 structures and nascent DNA and more G4 regions identified genomically
- 4) The ATRX/DAXX complex, including ATRX helicase activity and H3.3 chaperone activity, are required to protect cells from G4-mediated stress
- 5) Mechanistically, ATRX/DAXX-mediated H3.3 deposition functions upstream of heterochromatin formation, which is ultimately required to protect these regions
- 6) G4 regions identified experimentally in human cell lines show high mutation burden in both ATRX and ESET mutant cancers

Overall, we feel that our manuscript has greatly improved through the review process. Please find our response to specific comments below.

REVIEWER COMMENTS

Reviewer #1 (Remarks to the Author):

Several biological functions of ATRX have been described in detail. These include ATRX as a chromatin remodeller, as histone chaperone involved in the deposition of histone H3.3, it localizes to heterochromatin and acts as a helicase resolving G-quadruplex (G4) structures during replication. Mutations in ATRX leads to severe human diseases. The authors of this manuscript address questions related to these described functions of ATRX with a special focus on the relation between ATRX and G4 structures. They end up with a very nice model figure on the function of ATRX, but on the way to this model several problems of this manuscript have to be clarified. These include

Page 3/4 the data clearly show that ATRX colocalizes with G-rich and potential G4 forming sequences. But they do not tell whether ATRX indeed binds to G4 structures. Why are no ATRX/G4 stainings shown for G1 and G2 phase in Figure 1d. It would be expected that the number of G4 spots decrease during S phase but this is not clear from the figure. A statistical analysis the number of ATRX/G4 spots during all phases of the cell cycle would be needed. How does the staining look like in metaphase chromosomes?

We have now completed a more thorough assessment of ATRX-G4 colocalization throughout the cell cycle. We analyze cells each during G1 (n=47), early S (n=53), and late S (n=34). As the reviewer suggests we might, we observe a trend in which ATRX-G4 mean intensity decreases as cells enter S-phase, consistent with replication-coupled resolution of G4 structures (**Fig. 1e-f**). We do not observe strong ATRX-G4 association on metaphase chromosomes (**Supplementary Fig. 2c**). Additionally, we now include single antibody controls for these experiments, in addition to ATRX-G4 PLA staining performed in ATRX KO cells. These data are included as **Supplementary Fig. 3**.

Finally, before asking whether ATRX binds to G4 in vivo authors should show that it binds to various G4 structures in vitro.

The premise of our study is rooted in a strong body of cellular studies linking ATRX and G4 biology (Law et al. 2010; Watson et al. 2013; Clynes et al. 2015; Wang et al. 2019; Zyner et al. 2019). This includes seminal work demonstrating that ATRX can bind G4 DNA in vitro (Law et al. 2010). We agree that it would be interesting to address the breadth of various G4 structures that can be bound by ATRX and that this should be the focus of future studies. This point is now raised in the discussion.

Page 4: Minor: what means „replicative role“?

We have clarified this to “replication-associated role”. We refer to data from Clynes et al 2014 showing that loss of ATRX resulted in a prolonged S phase and an increase in replication fork stalling and DNA double strand breaks. Further, work from Clynes et al 2015 and Huh et al 2016 demonstrate that ATRX facilitates replication fork processivity.

Page 3-5: ATRX seems to interact with a large number of other proteins and most of these interactions are required for proper ATRX function. Have the authors made any attempt to identify which region of ATRX interacts with which other protein. And how are the interactions between ATRX interaction partners. These questions may not be essential for aims of this manuscript, but they should be at least mentioned.

We agree with the reviewer that more insight is needed into the molecular mechanisms by which ATRX interacts with its protein partners and that these experiments should be the focus of future studies. To start to address determinants of ATRX proximity to MCM, we now make more use of ATRX helicase mutants and ATRX mutants that disrupt DAXX binding throughout our study (see responses below). We find that ATRX maintains proximity to MCM both in the absence of its helicase activity and in the absence of its association with DAXX (**Fig. 2c** and **Supplementary Fig. 5**), suggesting that ATRX helicase and histone chaperone activity is decoupled from mechanisms that recruit ATRX to specific genomic regions. We also now point out in the discussion that direct molecular mechanisms remain to be explored.

Page 5/6: from the data presented it is not clear whether there is a direct interaction of ATRX with the MCM helicase or whether this interaction occurs via DAXX. Biochemical analyses would be required in addition to the presented immunoprecipitation experiments.

We have performed a number of experiments to address the question of DAXX dependence. First, we performed IP-WB of ATRX from DAXX KO cells, where we find that ATRX maintains association with MCM6 (**Supplementary Fig. 5**). In support, we find that an ATRX mutant that disrupts DAXX binding (L1238A) also maintains association with MCM6 (**Fig. 2c**). Additionally, we performed ATRX-MCM6 PLA assays in both DAXX KO cells and in ATRX KO cells expressing the L1238A mutant. Again, we observe ATRX-MCM6 proximity under both experimental settings, included as **Supplementary Fig. 5b-c** and **Supplementary Fig. 5e-f**. These results suggest that ATRX recruitment to certain replicating foci is independent of DAXX.

In addition, which helicase (ATRX or MCM) resolves G4 structure as claimed in Figure 4 (see also general comment).

We now report that ATRX maintains MCM6 interaction even when key residues in its helicase domain are mutated (see **Fig. 2c** and **Supplementary Fig. 5e-f**). Further ATRX helicase mutants remain sensitive to PDS (**Fig. 4b**), suggesting that the ATRX helicase plays a role in resolving G4 structures. Additionally, we now provide PLA staining of G4-EdU in ATRX KO cells

expressing ATRX helicase mutants. These data show that G4-EdU co-occurrence remains high in the absence of ATRX helicase activity and are included as **Supplementary Fig. 7**. While this experiment does not rule out that the MCM helicase plays a role in resolving G4 structures, it does implicate the ATRX helicase in this process.

Page 6: Authors claim that G4 structures accumulate at replication sites in the absence of ATRX. They conclude that the ATRX/DAXX complex is required to remove (resolve) G4 structures during early replication. This is supported by one microscopic figure. This is not acceptable for such a claim. Either many nuclei from ATRXO or DAXXO cells have to be shown or a panel of such nuclei should be presented. The G4 spots probably should represent G4 structures in replication factories. However, when looking at replication factories in the literature they are much smaller and more distinct. Authors should show G4 signals in normal replicating cells and subsequently they have to explain the strange looking G4 signals in their knock out cells.

We now provide additional cell images and quantification to support this conclusion. Data are quantified from 35-52 nuclei (**Fig. 3a** and **3b**). Further, we have now performed CUT&Tag G4-sequencing in WT and ATRX KO cells, demonstrating that the number of observed G4 regions increases in the absence of ATRX. These data are included as **Fig. 3** and **Supplementary Fig. 8** and are discussed in more detail below.

As a general observation, the EdU-G4 foci observed in WT cells are small and distinct. While many foci have similar appearance in the ATRX and DAXX KO cells, we do see more foci that are larger or have blurred boundaries. We do not have an experimental explanation for this, but perhaps this is indicative of the type of sequence that may be protected by ATRX (e.g., telomeres or repetitive regions). We will continue to think about this issue.

Pages 6-9: Figure 4 claims that ATRX/DAXX is required for removing G4 structure at sites of replication. In the following experiments evidence is provided that ATRX is involved in the deposition of H3.3 and maintenance/establishment of a heterochromatin at these sites. Authors mention that G4 structures form in nucleosome-free chromatin and that the establishment of heterochromatin may remove G4 structures. The model implies that ATRX is recruited to sites of some G4 structures and these are removed by the deposition of H3.3.

General comment: this manuscript contains some severe inconsistencies which have not only been explained but also experimentally addressed. Figure 4 shows that G4 structures accumulate in the absence of ATRX suggesting an active role of ATRX (or the MCM helicase) in resolving this structure. G4 preferentially forms in nucleosome free regions of the genome. But it is explained that ATRX maintains a heterochromatic structure at G4 sites. Therefore it could be that the effect described in Figure 4 is that since heterochromatin is no longer retained in the absence of ATRX G4 structure could now become formed. Or does the formation of heterochromatin by ATRX resolve G4 structures – a very novel mechanism. Experimental answers to these and the other question mentioned above have to be incorporated into a revised version.

This is a really interesting idea, that the histone chaperone function of ATRX and the role of ATRX in promoting H3K9me3 heterochromatin formation may uniquely contribute to the role of ATRX in G4 biology. To address this issue, we have performed a number of new experiments, using ATAC-seq to assess open chromatin at ATRX-bound G4 regions. First, we extended our ATAC-seq analysis to include ATRX KO cells expressing various ATRX mutants. These experiments show that both the helicase activity of ATRX and its interaction with DAXX (so

putative chaperone function) are required to “close” the ATRX-bound G4 regions (**Fig. 5e**). Second, we analyzed published ATAC-seq data from H3.3 KO cells expressing a mutant of H3.3 that interacts with DAXX but cannot be incorporated into chromatin (H3.3 L126A I130A). These data demonstrate that H3.3 must be deposited at ATRX-bound G4 regions to keep them “closed” (**Fig. 5f**). Both of these activities (helicase domain and chaperone activity) are also required for H3K9me3 at ATRX-bound G4 regions (**Fig. 6**). Finally, we analyzed our previous H3.3 ChIP-seq and performed new ATAC-seq from ESET KO cells, the histone methyltransferase responsible for H3K9me3 at ATRX-bound G4 regions. We find that while H3.3 deposition is not altered in the absence of ESET, its loss (along with H3K9me3 loss) does lead to increased accessibility at ATRX-bound G4 regions (these data are included in a new **Fig. 7**). Overall, these results are very exciting and in line with the reviewer’s suggestion that ATRX helicase activity, DAXX-mediated H3.3 deposition, and subsequent heterochromatin establishment all play a role in protecting cells from G4-mediated stress (see new model in **Fig. 9**).

In addition, most of the claims come from immunoprecipitation experiments or not really convincing immunofluorescence pictures. Additional biochemical methods should be used to support the claims.

We have improved the quality of our images, we apologize for the previous poor resolution. We also expand the scope of our manuscript to include ATRX mutants or H3.3 mutants in the majority of our assays. We agree that additional biochemical dissection of ATRX protein interactions is important, and this will be the focus of future studies, noted in the discussion.

In summary, this manuscript shows some nice observations but as it is in its present form it gives no more mechanistic explanations about the function of ATRX as already present in the literature.

We thank the reviewer for their suggestions and hope that the revised manuscript now provides improved insights into ATRX function in G4 biology.

Reviewer #2 (Remarks to the Author):

The ATRX-DAXX complex assembles the histone H3.3 variant into nucleosomes in heterochromatin regions of the genome. Loss of function mutations in ATRX and DAXX support a role for this complex in suppression of tumorigenesis. However, their functional role as possible tumour suppressors remains largely uncharacterised. ATRX contains ATPase activity of unknown function. Previous studies have found that ATRX deficiency sensitizes cells to small molecules that bind to and stabilize G-quadruplex structures, suggesting that ATRX may resolve these structures. However, the previous data showing interaction between ATRX and G-quadruplexes is generally of low quality, and whether ATRX directly interacts with these structures remains debated. Previous studies have also linked loss of ATRX to genomic instability, possibly caused by defects in DNA replication fork progression.

Here, Tang and colleagues explore the role of ATRX and DAXX in G-quadruplexes in mouse ES cells. The authors found that 40% of ATRX ChIP seq peaks are near sequences that have the propensity to form G-quadruplexes. The majority of ATRX peaks don’t overlap with predicted G-quad structures, but rather with what appears to be simple repeats. The authors found that loss of ATRX causes subtle reductions in H3K9me3 and H3.3 ChIP-seq signal at many putative G-quad sequences. ATRX transgenes that interfere with DAXX interaction and ATPase activity exhibit similar sensitivity as the ATRX knockout cells to a G-quadruplex

stabilizing molecule (pyridostatin), suggesting that ATRX may resolve these structures. Using co-IP experiments followed by mass spectrometry analysis to determine associating proteins, the authors found MCM proteins and the two subunits of the FACT histone chaperone complex. Knockdown of FACT subunits didn't affect ATRX-DAXX association with G4 structures by ChIP-seq, and the functional significance of ATRX-FACT interaction remains unknown. Using SNS-seq, the authors found DNA replication origins near some ATRX-binding sites. EdU labeling and sequencing analysis suggests that some origins may initiate replication earlier in the absence of ATRX.

The strength of this study is the use of high resolution ChIP seq to map ATRX-DAXX and H3.3 to regions of the genome that likely have the propensity to form G-quadruplexes. Unfortunately, the authors don't use the G-quadruplex antibody for ChIP-seq in their knockout ATRX, DAXX, and H3.3 cell lines to specifically address if loss of these proteins leads to increased G-quadruplex structures across the genome. Instead, they use the G-quad antibody in relatively low resolution proximity ligation assay in order to make conclusions about how ATRX-DAXX G-quadruplex structures and origin function (as measured by EdU incorporation). Previous studies have used the BG4 antibody to map G-quadruplex structures by next generation sequencing, and the authors need to determine if the loss of DAXX, ATRX, or ATRX ATPase activity affects the overall number of G-quadruplex structures. The PLA data is not convincing, and as I mention below various controls that are missing for these assays.

We appreciate the reviewer's comment regarding our strong genomics data. We have expanded on this data to include ATAC-seq analysis of ATRX mutant addbacks in ATRX KO ESCs and H3.3 mutant addbacks in H3.3 KO ESCs (**Fig. 5**). We also include ATAC-seq analysis of ESET KO ESCs, the HMT responsible for H3K9me3 at G4 regions in ESCs (**Fig. 7**). We also appreciate that many controls are needed for PLA, which we now include. Regarding the BG4 sequencing - we have found this genomics assay to be quite challenging. After many months of effort, we were able to generate high quality data sets mapping G4 regions in WT and ATRX KO ESCs, which we include in our resubmission (**Fig. 3** and **Supplementary Fig. 8**). We discuss these results in more detail below.

Numerous biochemical and genetic studies over the past twenty some years have demonstrated that the active MCM helicase involved in DNA replication origin firing consists of MCM2,3,4,5,6,7. However, only MCM2,4,6,7 are mentioned as interacting with ATRX. It is known that sub complexes of the MCM proteins exist, and some biochemically abundant sub complexes of unknown function (such as MCM2,4,6,7) can occur outside of the origin-competent full MCM hexamer consisting of MCM2-7. The authors finding (fig S4D) that ATRX does not interact with MCM3,5 really calls into question the model presented, namely that ATRX-DAXX interacts with the MCM proteins at origins of replication.

It is true that we did not observe association of ATRX with MCM3, 5 in our pull-down of ATRX from ammonium sulfate nuclear lysates. We wondered if this might be a technical issue with lysate prep. To determine whether ATRX is in close proximity to MCM3, 5 in cells, we performed PLA assays including quantification in both WT and ATRX KO cells. These data are included as **Supplementary Fig. 4d-k** and show that ATRX is in fact in close proximity to MCM2-6 in cells (including MCM3, 5). For clarity, we have removed the IP-WB data for MCM3, 5.

Moreover, the finding of little EdU incorporation near sites of ATRX and MCM (Fig 1b) also suggests that ATRX is not present at active origins of replication. The SNS-seq data do suggest

that some origins are located near some ATRX binding sites, but additional analyses that quantitate these findings may help shed light on the functional significance of these findings.

As stated, we see little overlap of ATRX-MCM and EdU in relatively low resolution imaging studies. Based on the small number of ATRX enriched regions we observe (<500) compared to the very large number of replication origins per cell, we do not expect every MCM complex to interact with ATRX. As suggested, we have performed additional analysis related to SNS-seq. Specifically, we find that origins stabilized by PDS treatment show increased statistical likelihood of proximity to ATRX-bound G4 regions. These data are included as **Supplementary Fig. 9a-d**. However, when considering the whole of experimentally identified BG4 peaks in the absence of ATRX, we do not observe any evidence of increased origin usage from these regions (**Supplementary Fig. 9g**), while we do observe increased levels of newly synthesized DNA from these regions in the absence of ATRX or DAXX (**Fig. 3g**). This technique, EdU-seq, cannot distinguish between replication and damage repair, and will capture both events. Overall, we feel that these analyses support a function for ATRX in preventing G4-associated DNA damage rather than G4-associated origin firing (although some origins may be affected by loss of ATRX). We sincerely thank the reviewer for suggesting this analysis.

Because the authors don't know how ATRX-DAXX interacts with the MCMs, experiments that could tease out the functional significance of this interaction remains unclear. As I wrote above, the evidence presented that ATRX-DAXX impacts origin function is currently weak. However, I think the manuscript could be greatly improved if the authors should pursue high resolution ChIP seq experiments to test their model the necessity of the ATRX-DAXX complex and ATPase activity for resolving G-quadruplex structures in vivo. Without additional experiments, the findings regarding ATRX-DAXX resolution of G-quadruplexes remains undeveloped and the study represents a very limited advance in our understanding of ATRX-DAXX complex function. These issues and other issues noted below should be resolved prior to consideration of acceptance of this manuscript.

We agree with the reviewer that high-resolution ChIP experiments offer many advantages over lower-resolution microscopy assays. With great effort (really, this was very difficult), we now have obtained high-quality G4 CUT&Tag datasets in WT and ATRX KO cells (**Fig. 3** and **Supplementary Fig. 8**). The most important message is that the number of observed G4 regions increases in the absence of ATRX. Specifically, we find 6,301 novel G4 sites in ATRX KO ESCs that are not observed in wild-type ESCs (**Fig. 3d**). These regions are annotated predominantly as introns, intergenic regions, LINEs, and LTRs (**Fig. 3f**), suggesting that ATRX is mainly responsible for resolving G4 structures in these regions. As the reviewer suggested, we also tried to map G4 structures in our ATRX addback cell lines to determine whether helicase and chaperone activity are required. Unfortunately, we did not observe enrichment over input from these samples. We are happy to provide this negative data for reviewer assessment should they wish to see it.

In addition, we now provide H3K9me3 ChIP-qPCR and ATAC-seq demonstrating that both the ATRX helicase domain and its ability to interact with DAXX are necessary to maintain chromatin and more specifically heterochromatin at ATRX-bound G4 regions (**Fig. 5** and **Fig. 6**). Further, we show that ESET KO ESCs show loss of heterochromatin and increased levels of open chromatin at G4 regions, all while maintaining H3.3 deposition (**Fig. 7**). Interestingly, we find that ESET KO cells also show sensitivity to PDS (**Fig. 7e**). Overall, these data support our improved model (**Fig. 9**), in which ATRX helicase and chaperone function are upstream of heterochromatin formation at ATRX-bound G4 regions, which ultimately protects cells from G4-mediated replicative stress.

Other comments and issues that should be resolved:

1) The authors found an enrichment of predicted G-quadruplex structures near 40% of ATRX ChIP seq peaks. The majority of ATRX peaks don't overlap with predicted G-quad structures, but rather with what appears to be simple repeats. What chromatin features are found at these simple repeats that seem to promote ATRX binding? Previous studies have found that the ATRX ADD domain has high affinity for H3K9me3, and the authors genome browser screenshots show H3K9me3 at predicted G4-quadruplex structures. Genome-wide, what is the overlap between H3K9me3 and the predicted G-quadruplex structures? Is H3K9me3, rather than G-quadruplex, a predictor of ATRX ChIP seq peaks? The authors should perform analyses comparing the ChIP seq data sets.

The question of ATRX recruitment is a very important one. We have now subjected our ATRX-bound regions to a 12-state ChromHMM analysis (pipeline generated using H3K4me1, H3K4me3, H3K9ac, H3K27ac, H3K36me3, H3K27me3, H3K9me3, CTCF, Oct4, and Nanog). We find that ATRX-enriched regions were generally assigned to the "intergenic" designation that is characterized by lack of chromatin signature ($n = 272/435$). This is also the case for the ATRX-bound simple repeats designated by HOMER. In agreement with the known relationship between ATRX and H3K9me3, the next most highly enriched term is "heterochromatin" as defined by H3K9me3 ($n = 53/435$). However, our previous assessment of H3.3 and H3K9me3 in ESCs (Elsässer et al. 2015) demonstrate that there are a large number of H3K9me3 enriched regions that do not incorporate H3.3, suggesting that H3K9me3 cannot be the sole determinant of ATRX/DAXX recruitment to chromatin. To further this analysis, we have performed G4 motif prediction on all H3K9me3 bound regions in ESCs. Only a small number of H3K9me3 peaks are predicted to contain G4 structures, and we do not find any enrichment in the number of predicted G4 regions over iteratively selected control regions (see reviewer figure below). While we cannot state with certainty how ATRX is recruited to target regions, we hope to more clearly delineate these relationships in future studies.

A. G4 prediction of H3K9me3 enriched regions in ESCs. **B.** Number of predicted G4 motifs in H3K9me3 peaks ($n=2,717$) compared with the number of G4s expected in a randomly and iteratively selected group of the same size (avg=6,224)

2) The authors model- that ATRX-DAXX interacts with MCMs at origins of replication also predicts that H3.3 should be enriched at these origins. Is H3.3 preferentially found at origins of

replication? Do MCM proteins ChIP to G-quad or other genomic regions where ATRX ChIPs? For example, the genomic regions found in the SNS-seq analysis.

Our model would predict that a very small subset of origins may have H3.3 deposition, although this interpretation is complicated by the fact that genic H3.3 deposition is strongly associated with H3K4me3 / H3K27ac / open chromatin, all markers associated with replication activation. We do see that H3.3 is enriched at ATRX-bound G4 regions, requiring both ATRX and DAXX for deposition (**Supplementary Fig. 10c, 10d**). We have not been able to obtain our own MCM ChIP. We presume that MCM enrichment on chromatin would be extensive, at both active origins and those that do not fire.

3) The authors use SNS-seq to identify origins of replication near ATRX-enriched G-quadruplexes. Figure 6d explores possible origins near ATRX-enriched G-quads, however, it's unclear how many 'ATRX-bound G-quad' genomic regions were used for this analysis. What proportion of total G-quad sequences are found in the SNS-seq data? This type of analysis would be important in order to determine putative origins are near G-quad sequences and ATRX binding sites.

All ATRX-bound G4 regions were used for the analysis of SNS-seq data included in the original submission. As with MCM, we expect that the number of origins identified by SNS-seq will far exceed the number of origins that may be influenced by ATRX activity. We have now determined proximity of ATRX-bound G4 regions to origins identified by SNS-seq. Under normal conditions, we find that ~25% of ATRX-bound G4 regions are within 5 kb of an origin. This number increases when origins are identified from PhenDC3 treated samples, in which G4 structures are stabilized, where now ~50% of ATRX-bound G4 regions are within 5 kb of an origin. The observed overlap between ATRX enriched regions and origins is statistically significant when the peak is predicted to contain a G4 motif. These data are included as **Supplementary Fig. 9a-d**.

4) The authors should provide screen shots of EdU seq and SNS seq at putative origins in WT and ATRX delete cells.

We now provide browser tracks of EdU-seq in WT and ATRX KO cells and SNS-seq representative of putative origins in WT and PhenDC3-treated cells, included as **Supplementary Fig. 9a**.

5) Previous studies have found that MCM2 has histone H3-binding activity, and it remains unclear from the experiments presented here that the MCM2,4,6,7 association with ATRX-DAXX isn't simply through bridging via interaction with H3.3-H4. The authors should determine if ATRX and DAXX can still associate with the MCM proteins in the absence of H3.3.

This is an interesting and thought-provoking question. We performed the suggested experiment, IP of ATRX in the absence of H3.3 with WB for MCM proteins. We find that ATRX-MCM association is reduced in the absence of H3.3. These data are included as **Supplementary Fig. 6b**. Similar results are observed from H3.3 KO cells expressing an H3.3 mutant that is not stably incorporated into chromatin (H3.3 L126A I130A) - see **Fig. 2d**. Overall, these data suggest that H3.3 may be a substrate for both the ATRX-DAXX complex and the MCM complex. While we cannot support this conclusion with the data at hand, we would be very excited if ATRX/DAXX were to deposit H3.3 at G4s upstream of the MCM helicase, and MCM2 would then recycle this H3.3 back into chromatin during replication.

6) The PLA lacks important controls that are needed in order to assess the signal from these experiments. For example, the control assays should be conducted with only one of the antibodies. Additionally, the assay should be performed in ATRX deficient cells in order to control for possible ligation artifacts caused by non-specific antibody interactions.

We now include these important controls. Single antibody PLA are included as **Supplementary Fig. 3**. Critical PLA assays using ATRX antibody are performed in ATRX KO cells (see **Supplementary Fig. 3a, Supplementary Fig. 4d, 4f, 4h, 4j** and **Supplementary Fig. 5b**).

Based solely on the evidence provided in the manuscript, it's not clear that there is a substantial overlap between ATRX and EdU staining in late S phase. Thus, quantitation of this assay is required in order to determine the statistical significance of this result.

Given the number of active origins in a replicating cell and the number of ATRX-bound G4 regions we identify, we don't expect to see substantial overlap between ATRX and Edu staining. However, we do see increased G4-EdU in both ATRX KO and DAXX KO ESCs (**Fig. 3a** and **3b**), and we now show that both the helicase and DAXX interaction are required to rescue this effect in ATRX KO ESCs (**Supplementary Fig. 7**).

7) Figure 2A pyridostatin assay of ESCs needs to include IC50 values for the different genotypes.

Nonlinear regression assay and IC50 are now included for WT (0.8042 μ M), ATRX KO (0.3962 μ M), DAXX KO (0.3837 μ M), and HIRA KO (0.9088 μ M).

Reviewer #3 (Remarks to the Author):

In this manuscript, Teng et al address ATRX function and mechanism at G4 structures. They show that ATRX is in close proximity to G4 structures in vivo and that this depends on the DAXX binding and helicase activity of ATRX. They identify MCM and FACT as ATRX/DAXX interactors and that ATRX functions to maintain G4 chromatin in a closed state and that loss of ATRX results in G4 accumulation at DNA replication forks. Some of the results presented are expected, others are intriguing. The authors make some overstatements and some of the experiments lack critical controls. However, with significant revisions, this manuscript can be considered for publication.

1. Fig 1 - The authors state that ATRX and G4 associate in all phases of the cell cycle. Can they show a cell cycle phase where there is not much interaction eg – G2/M, where G4 levels should be low due to lack of DNA replication?

We now provide more complete assessment of ATRX-G4 interaction throughout the cell cycle, see also response to R1. By trend, we see highest ATRX-G4 association in G1 with reduced signal intensity from the PLA assay in early S phase, consistent with G4 resolution as part of the replication process. We do not see a great deal of ATRX-G4 foci on mitotic chromosomes. We rephrase the text to reflect this result. These data are now included as **Fig. 1e-f** and **Supplementary Fig. 2c**.

The authors have to show lack of associated between another unrelated chromatin bound protein (HIRA?) and G4 to show that this association is specific to ATRX.

This is an important control, we now include HIRA-G4 PLA, showing no co-localization and supporting a specific ATRX-G4 interaction. To validate that the HIRA antibody is working in this experiment, we include PLA confirming a previously reported co-localization between HIRA and RNAPII (Ray-Gallet et al. 2011), including single antibody controls. These data (both single channel and merge) are included as **Supplementary Fig. 3b-c**, including image quantification.

2. Fig 1- The treatment with DNaseI abolishes PLA signal. Does it abolish interaction signal between two proteins eg. ATRX and DAXX? Does DNase treatment abolish EdU signal (since this appears to be done only in G1 samples). This is required to show the specificity of DNase treatment to protein-G4 structures and not a non-specific effect on all PLA derived signal. All PLA images must be quantified (Figs. 1e, 3c, 3f to show number of EdU positive and PLA signal positive cells). Show different channels for IF. The merged images are not easy to assess.

This is a good point that the PLA assay may generally be disrupted by DNaseI treatment, since DNaseI may be difficult to wash out and then subsequently may digest DNA primers from the PLA antibodies. Indeed, we think this is the case, as DNaseI treatment does reduce ATRX-DAXX interaction observed by PLA but not by IP-WB (see reviewer figure below). Our IP-WB is in agreement with published results that the ATRX-DAXX interaction does not require DNA. Therefore, DNaseI is not an appropriate control for a PLA assay. We now remove this experiment from our submission. In place, we now provide all single antibody stains and experiments performed using ATRX KO cells as control. Additionally, we now provide all PLA data as both single channel and merge images. Further, we now quantify all PLA imaging as suggested.

A. PLA of ATRX-DAXX when slides were untreated (top) or treated with DNase I (bottom) shows that signal is reduced with DNase treatment. **B.** Quantification of images in panel A. **C.** IP of ATRX followed by WB for ATRX and DAXX in lysates treated as indicated. Experiment shows that ATRX-DAXX interaction is not disrupted upon DNase treatment. We believe that reduced

signal in PLA assay is an artifact due to residual DNase activity on the DNA-hybridized secondary antibodies.

3. Pg 4 – “Further, mutations to ATRX that disrupt its helicase activity (K1562R, K1612N) remain sensitive to PDS, in support of the hypothesis that ATRX unwinds G4 structures in vivo (Fig. 2b and Supplementary Fig. 3d)”. This is an overstatement. There is no evidence in this manuscript that ATRX is able to unwind G4 structures. It could inhibit G4 formation or recruit another helicase that does the unwinding. This conclusion must be rephrased.

This overstatement has been removed.

4. Supplementary Fig 4d – show ATRX western. Also show DAXX western to show that ATRC can IP a known interactor when it cannot IP mcm3 and mcm5. Supplementary Fig 4e – show DAXX western to confirm that DAXX is being IPed. Also, as the authors stated, DAXX binds numerous proteins in the cell independent of ATRX. One of these should be used as a control in DAXX IPs.

We have added appropriate controls to supplemental figures.

5. Both MCM6 and FACT are abundant proteins. I am not sure that PLA with these proteins and another chromatin bound protein (ATRX) is even meaningful. Can the authors show another chromatin bound protein that does interact with FACT components, but interacts with its cognate interactor?

This is a good point that MCM and ATRX may appear to be in close proximity simply because of the abundance of MCM on chromatin. However, based on immunoprecipitation, we do not observe interaction between MCM and HIRA (**Supplementary Fig. 5g**). Further we do not see HIRA proximity to G4 based on PLA, while we do see HIRA interaction with RNAPII as previously reported (Ray-Gallet et al. 2011) (see **Supplementary Fig. 3b-c**). These data serve as powerful internal controls in support of proximity between ATRX and MCM.

6. DAXX binding mutant L1238A and Helicase mutant K1562R, K1612N are sensitive to PDS. Does treatment of cells with PDS increase G4 to the same extent? Can the BG4 antibody be used for a dot plot analysis and levels of G4 quantified?

This is an excellent question. Based on reviewer suggestion, we now provide CUT&Tag using the BG4 antibody in WT and ATRX KO ESCs, discussed below. As mentioned previously, this experiment was technically very challenging. Unfortunately, we cannot provide BG4 CUT&Tag for our ATRX mutant addbacks.

7. “Loss of ATRX leads to accumulation of G4 at replication sites” – This entire section would benefit from genomics or significantly higher resolution microscopy techniques. PLA experiments alone are not convincing. To state that G4 structures accumulate at replication sites, a detailed comparison of G4 ChIP (using BG4 or other) and some replication complex protein must be shown. Does MCM complex KD also result in G4 accumulation and/or ATRX recruitment as the model would suggest?

We agree that this conclusion would greatly benefit from genomics. With much effort we were able to obtain G4 CUT&Tag data sets from WT and ATRX KO cells (**Fig. 3** and **Supplementary Fig. 8**). These data support our conclusion that G4 occurrence is increased in the absence of ATRX. The most important message is that the number of observed G4 regions increases in the

absence of ATRX. Specifically, we find 6,301 novel G4 sites in ATRX KO ESCs that are not observed in wild-type ESCs (**Fig. 3d**). These regions are annotated predominantly as introns, intergenic regions, LINEs, and LTRs (**Fig. 3f**), suggesting that ATRX is mainly responsible for resolving G4 structures in these regions. Additionally, we observe increased EdU-seq signal from these regions in ATRX or DAXX KO cells, but we do not see increased origin activity at these regions in PhenDC3 treated cells (**Fig. 3g** and **Supplementary Fig. 9g**). Perhaps the EdU-seq signal that we observe is a result of DNA damage repair. As the reviewer suggested, we also tried to map G4 structures in our ATRX addback cell lines to determine whether helicase and chaperone activity are required. Unfortunately, we did not observe enrichment over input from these samples. We are happy to provide this negative data for reviewer assessment should they wish to see it.

Given the likely pleiotropic effects of MCM complex KD, we have not pursued this line of experiments.

8. Pg 6 - 'Our identification of an association between ATRX and the MCM and FACT complexes add a direct link to the existing literature suggesting that ATRX plays a role in DNA replication'. I think there is enough literature showing (and not merely suggesting) that ATRX loss does play a role in DNA replication.

We have updated this statement to be more reflective of the current literature.

9. Supplementary Fig 6 – Quality of PLA images are not convincing and need to be more clear.

We now include deconvoluted images with greater resolution for all PLA.

10. Fig. 5 – The ATAC Seq data is very compelling here and clearly shows increased accessibility at sites that contain G4 and ATRX. Can this be rescued by overexpression of DAXX (or H3.3) that is known to deposit variant histones (CENPA) non-specifically when it is overexpressed.

We are so intrigued by this ATAC-seq data, especially given our previously published observations that loss of H3.3, a marker of dynamic chromatin, has no effect on the open state of active enhancers in ESCs (Martire et al. 2019). Our observation here is in agreement with recent work from the Elsasser lab showing that H3.3 supports closed chromatin states at ERVs in ESCs (Navarro et al. 2020). We now expand this section of the manuscript in several ways. First, we now provide ATAC-seq analysis of ATRX KO cells expressing ATRX mutants used in our PDS assay. Here, we find that both ATRX helicase activity and interaction with DAXX (so putative chaperone activity) are required to maintain ATRX-bound G4 regions in a closed state (**Fig. 5e**). Second, we use published data from Navarro et al 2020 to analyze ATAC-seq at ATRX-bound G4 regions in H3.3 KO cells expressing an H3.3 mutant that binds to DAXX but cannot be incorporated into chromatin (H3.3 L126A I130A). Importantly, we find that H3.3 deposition is required to “close” the chromatin at these regions (**Fig. 5f**). Further, we perform the experiment suggested above by our reviewer. We overexpress DAXX-Flag in ATRX KO to determine whether this might result in improved H3.3 deposition, or perhaps CENPA deposition, at these regions which might rescue the phenotype observed. However, we find that DAXX OE in ATRX KO cells is not sufficient to “close” these regions of chromatin. We present these data here for reviewer evaluation.

A. WB of DAXX overexpression (OE) in ATRX KO cells. **B.** Box plots representing ATAC-seq read counts at ATRX-enriched G4 and non-G4 regions in ATRX KO ESCs either exogenously expressing vector or DAXX-Flag constructs. The bottom and the top of the boxes correspond to the 25th and 75th percentiles, and the internal band is the 50th percentile (median). The plot whiskers correspond to 1.5 interquartile range. Statistical significance is determined by Wilcoxon Mann Whitney test. ns, not significant.

11. The authors suggest that ATRX/DAXX does not contribute uniformly to transcriptional activity of genes containing ATRX enriched G4 regions. The authors should consider that their RNA seq looks at steady state levels and not rates of transcription. Since ATRX has been shown previously to facilitate elongation through structured DNA, it is possible that changes will only visible using methods such as Gro-seq or Pro-seq. This may be worth discussing further.

We have unpublished GRO-seq data from WT and H3.3 KO cells, which we have now analyzed at ATRX-bound G4 regions. We do not see any statistical difference in nascent transcription at these regions in the absence of H3.3, nor any change in nascent transcription from nearest neighboring genes. We include these analyses for reviewer assessment. While we have not carried out these experiments in chaperone KO cells, given our data suggesting an important role for ATRX/DAXX chaperone function at these regions, and the overall phenocopy of ATRX, DAXX, and H3.3 KO cells in our assays, we predict that these results would hold true for ATRX KO cells, as well.

A. GRO-seq analysis of signal from nearest neighboring genes (NNG) of ATRX peaks in WT and H3.3 KO ESCs. **B.** GRO-seq analysis of signal from ATRX-enriched regions in WT and H3.3 KO ESCs. n.s. not significant.

12. Fig 6 – Can levels of G4 be tested by BG4 ChIP next to some of the replication origins in WT and ATRX KO? Do G4 levels go up?

Yes, using G4 CUT&Tag, we do find that G4 levels increase at enriched regions in the absence of ATRX. We also find an increase in the number of enriched G4 regions in the absence of ATRX. These data are now included as **Fig. 3** and **Supplementary Fig. 8** and described in more detail above.

- Clynes, David, Clare Jelinska, Barbara Xella, Helena Ayyub, Caroline Scott, Matthew Mitson, Stephen Taylor, Douglas R. Higgs, and Richard J. Gibbons. 2015. "Suppression of the Alternative Lengthening of Telomere Pathway by the Chromatin Remodelling Factor ATRX." *Nature Communications* 6 (July): 7538.
- Elsässer, Simon J., Kyung-Min Noh, Nichole Diaz, C. David Allis, and Laura A. Banaszynski. 2015. "Histone H3.3 Is Required for Endogenous Retroviral Element Silencing in Embryonic Stem Cells." *Nature* 522 (7555): 240–44.
- Law, Martin J., Karen M. Lower, Hsiao P. J. Voon, Jim R. Hughes, David Garrick, Vip Viprakasit, Matthew Mitson, et al. 2010. "ATR-X Syndrome Protein Targets Tandem Repeats and Influences Allele-Specific Expression in a Size-Dependent Manner." *Cell* 143 (3): 367–78.
- Martire, Sara, Aishwarya A. Gogate, Amanda Whitmill, Amanuel Tafessu, Jennifer Nguyen, Yu-Ching Teng, Melodi Tastemel, and Laura A. Banaszynski. 2019. "Phosphorylation of Histone H3.3 at Serine 31 Promotes p300 Activity and Enhancer Acetylation." *Nature Genetics* 51 (6): 941–46.
- Navarro, Carmen, Jing Lyu, Anna-Maria Katsori, Rozina Caridha, and Simon J. Elsässer. 2020. "An Embryonic Stem Cell-Specific Heterochromatin State Promotes Core Histone Exchange in the Absence of DNA Accessibility." *Nature Communications* 11 (1): 5095.
- Ray-Gallet, Dominique, Adam Woolfe, Isabelle Vassias, Céline Pellentz, Nicolas Lacoste, Aastha Puri, David C. Schultz, et al. 2011. "Dynamics of Histone H3 Deposition in Vivo Reveal a Nucleosome Gap-Filling Mechanism for H3.3 to Maintain Chromatin Integrity." *Molecular Cell* 44 (6): 928–41.
- Wang, Yuxiang, Jie Yang, Aaron T. Wild, Wei H. Wu, Rachna Shah, Carla Danussi, Gregory J. Riggins, et al. 2019. "G-Quadruplex DNA Drives Genomic Instability and Represents a Targetable Molecular Abnormality in ATRX-Deficient Malignant Glioma." *Nature Communications* 10 (1): 943.
- Watson, L. Ashley, L. Ashley Watson, Lauren A. Solomon, Jennifer Ruizhe Li, Yan Jiang, Matthew Edwards, Kazuo Shin-ya, Frank Beier, and Nathalie G. Bérubé. 2013. "Atrx Deficiency Induces Telomere Dysfunction, Endocrine Defects, and Reduced Life Span." *Journal of Clinical Investigation*. <https://doi.org/10.1172/jci65634>.
- Zyner, Katherine G., Darcie S. Mulhearn, Santosh Adhikari, Sergio Martínez Cuesta, Marco Di Antonio, Nicolas Erard, Gregory J. Hannon, David Tannahill, and Shankar Balasubramanian. 2019. "Genetic Interactions of G-Quadruplexes in Humans." *eLife* 8 (July). <https://doi.org/10.7554/eLife.46793>.

REVIEWERS' COMMENTS

Reviewer #1 (Remarks to the Author):

All reviewers suggested that this manuscript may be acceptable for publication after extensive revisions and additional experiments. Authors now carefully addressed the majority of concerns. Although not all questions could be answered in this revised version the manuscript is greatly improved and even a new mechanistic model about ATRX function is provided.

Reviewer #2 (Remarks to the Author):

The revised manuscript contains several new controls and important experiments. Specifically, the addition of G-quad CHIP seq data in figure 3 from both WT and ATRX mutant cells is important and increases the overall quality of the manuscript. Specifically, these data support one of the major themes of this manuscript— namely, that ATRX associates with genomic regions that have the propensity to form G-quadruplex structures and that loss of ATRX results in an increased number of these structures. Ideally, the study would use rescue transgenes containing ATRX or a catalytically dead ATRX transgene to determine if the ATPase activity is necessary for changing the number of G-quad structures (as measured by CHIP with the BG4 antibody).

However, a main conclusion of the study — specifically, the association of ATRX with the MCMs remains poorly supported. Previously, I noted that the absence of MCM3 and MCM5 in the ATRX co-IP material was worrisome because these proteins are necessary for the physiologically relevant MCM complex that is involved with S-phase DNA replication. In the revised manuscript, the authors have not provided convincing evidence that the interaction between ATRX and the MCM proteins does not occur through contaminating nucleic acid. The authors find MCM2,4,6,7 and the subunits of the MRN complex in their ATRX co-IP data. One thing that both the MCMs and MRN complexes have in common is that they are topologically linked to DNA, and thus are often contaminants in IPs of proteins that bind to DNA/RNA. Previous studies have found that ATRX interacts with both RNA and DNA, so the possibility that nucleic acid is bridging the supposed association between ATRX and the MCMs is high.

In the revised manuscript, the authors have removed the 'negative data' for the ATRX association with MCM3 and MCM5. This omission is a mistake, and I think that it's important to keep these data in the manuscript in order for readers to understand that ATRX does not associate with the full MCM2-7 complex. The physiological relevance of MCM2,4,6,7 subcomplex remains unclear, and I also think that the manuscript would benefit from a brief discussion of the finding that ATRX seems to associate with this particular form of the MCMs.

Reviewer #3 (Remarks to the Author):

This revised manuscript from Teng et al investigating mechanisms of ATRX at G quadruplexes is much improved from the original submission. The addition of the BG4 CUT&TAG data is very nice and strengthens the manuscript considerably. The authors have adequately addressed all my initial concerns and I recommend this manuscript for publication in Nature communications.

REVIEWERS' COMMENTS

Reviewer #1 (Remarks to the Author):

All reviewers suggested that this manuscript may be acceptable for publication after extensive revisions and additional experiments. Authors now carefully addressed the majority of concerns. Although not all questions could be answered in this revised version the manuscript is greatly improved and even a new mechanistic model about ATRX function is provided.

Reviewer #2 (Remarks to the Author):

The revised manuscript contains several new controls and important experiments. Specifically, the addition of G-quad ChIP seq data in figure 3 from both WT and ATRX mutant cells is important and increases the overall quality of the manuscript. Specifically, these data support one of the major themes of this manuscript— namely, that ATRX associates with genomic regions that have the propensity to form G-quadruplex structures and that loss of ATRX results in an increased number of these structures. Ideally, the study would use rescue transgenes containing ATRX or a catalytically dead ATRX transgene to determine if the ATPase activity is necessary for changing the number of G-quad structures (as measured by ChIP with the BG4 antibody).

We agree with the reviewer that the transgene experiments would greatly add to this manuscript. We did sincerely try to obtain these datasets, but, unfortunately, after several attempts, we did not obtain signal above background for our addback cell lines in the G4 CUT&Tag assay. We hope to optimize this protocol in the future.

However, a main conclusion of the study — specifically, the association of ATRX with the MCMs remains poorly supported. Previously, I noted that the absence of MCM3 and MCM5 in the ATRX co-IP material was worrisome because these proteins are necessary for the physiologically relevant MCM complex that is involved with S-phase DNA replication. In the revised manuscript, the authors have not provided convincing evidence that the interaction between ATRX and the MCM proteins does not occur through contaminating nucleic acid. The authors find MCM2,4,6,7 and the subunits of the MRN complex in their ATRX co-IP data. One thing that both the MCMs and MRN complexes have in common is that they are topologically linked to DNA, and thus are often contaminants in IPs of proteins that bind to DNA/RNA. Previous studies have found that ATRX interacts with both RNA and DNA, so the possibility that nucleic acid is bridging the supposed association between ATRX and the MCMs is high.

It's good to be skeptical of the MCM pulldown. To assuage these concerns, we have shown that (1) the ATRX antibody does not pull down MCM in ATRX KO lysates, and (2) HIRA, another H3.3 chaperone, does not interact with MCM. Regarding the nature of the interaction, our model is agnostic to the way in which ATRX and MCM interact. We simply provide additional evidence to support what is in the literature, that ATRX is molecularly important at a subset of newly synthesized DNA. Regardless, please consider that our nuclear extracts are made by stripping soluble proteins off of chromatin with ammonium sulfate. We then pellet the chromatin, which remains intact without a shearing step, using

ultracentrifugation to clarify the lysate that we use for immunoprecipitation. Despite the clarification, it is formally possible that our lysates are contaminated with DNA and contain other nucleic acids. To test whether these nucleic acids are bridging the ATRX-MCM interaction non-specifically, we have performed an additional ATRX immunoprecipitation in which lysates were (1) untreated, (2) treated with Ethidium Bromide, or (3) treated with RNaseA. Regardless of treatment, we observe ATRX-Mcm6 interaction. We include these data as Supplemental Fig. 4d and here for additional review.

In the revised manuscript, the authors have removed the 'negative data' for the ATRX association with MCM3 and MCM5. This omission is a mistake, and I think that it's important to keep these data in the manuscript in order for readers to understand that ATRX does not associate with the full MCM2-7 complex. The physiological relevance of MCM2,4,6,7 subcomplex remains unclear, and I also think that the manuscript would benefit from a brief discussion of the finding that ATRX seems to associate with this particular form of the MCMs.

Given that we cannot satisfactorily conclude the nature of the MCM complex with which ATRX interacts, we wish to leave our data and discussion as is. Peer review materials will be publicly available so that the community is aware of the points raised by the reviewer.

Reviewer #3 (Remarks to the Author):

This revised manuscript from Teng et al investigating mechanisms of ATRX at G quadruplexes is much improved from the original submission. The addition of the BG4 CUT&TAG data is very nice and strengthens the manuscript considerably. The authors have adequately addressed all my initial concerns and I recommend this manuscript for publication in Nature communications.

Thank you to all of the reviewers for making our manuscript better!